# Cropland expansion in the United States produces marginal yields at high costs to wildlife

Tyler J. Lark [1,2 ✉], Seth A. Spawn [1,2,3], Matthew Bougie[1,2] & Holly K. Gibbs[1,2,3]

Recent expansion of croplands in the United States has caused widespread conversion of grasslands and other ecosystems with largely unknown consequences for agricultural production and the environment. Here we assess annual land use change 2008–16 and its impacts on crop yields and wildlife habitat. We find that croplands have expanded at a rate of over one million acres per year, and that 69.5% of new cropland areas produced yields below the national average, with a mean yield deficit of 6.5%. Observed conversion infringed upon high-quality habitat that, relative to unconverted land, had provided over three times higher milkweed stem densities in the Monarch butterfly Midwest summer breeding range and 37% more nesting opportunities per acre for waterfowl in the Prairie Pothole Region of the Northern Great Plains. Our findings demonstrate a pervasive pattern of encroachment into areas that are increasingly marginal for production, but highly significant for wildlife, and suggest that such tradeoffs may be further amplified by future cropland expansion.

[1] Center for Sustainability and the Global Environment (SAGE), Nelson Institute for Environmental Studies, University of Wisconsin-Madison, 1710 University Ave, Madison, WI 53726, USA. [2] DOE Great Lakes Bioenergy Research Center, University of Wisconsin-Madison, Madison, WI, USA. [3] Department of Geography, University of Wisconsin-Madison, Madison, WI, USA. ✉email: lark@wisc.edu

Several studies and federal reports have documented a resurgence in conversion of grasslands and other natural and semi-natural areas to row-crop production in the United States (US) beginning in the mid-to-late 2000s[1,2]. The initial timing of this cropland expansion (~2007–12) coincided with periods of high commodity prices, rapid buildout of the biofuels industry, and reductions to the extent of federal land conservation programs—all conditions that have since subsided. The characteristics and persistence of expansion, however, remain highly uncertain, which impedes evaluation and formation of federal farm, energy, and conservation policies. The impacts of recent land conversion on both agricultural production and natural habitat are also largely unknown. These information gaps limit our ability to compare the consequences of cropland expansion around the world or against other means of increasing production such as agricultural intensification, thereby clouding navigation of the intertwined global challenges of improving food and fuel production while maintaining the integrity of ecosystems.

The US contains some of the most productive soils in the world and supports an immense proportion of global grain production[3,4]. However, further expansion is likely to embody several heightened trade-offs. For example, recent field scale analyses reveal globally significant carbon emissions from cropland expansion in the US[5,6], thereby reducing the climate benefits of expanding agriculture there relative to other regions such as the tropics[7,8]. New croplands in the US also tend to occupy areas with marginal biophysical characteristics such as erosive soils, poor drainage, nutrient or moisture deficiencies, or climatic stress[9]. While these limitations could constrain crop yields and diminish the returns from expansion, the magnitude of their impact is unknown. This uncertainty inhibits accurate assessment of the costs and benefits of further expansion.

Expansion in the US also threatens grasslands and other natural habitats that have high conservation value[10]. A recent United Nations report identified agricultural land use change as a primary driver of global biodiversity loss[11,12], but detailed analyses of crop expansion impacts in the US have not yet been conducted.

Previous studies have shown that grasslands are the primary source of land converted to crops in the US[9], and that this type of change can be particularly detrimental for many pollinators, birds, and the plant species upon which they rely[13–15]. Compared with croplands, for example, the typical grassland harbors more than 60 times as many milkweed pods—the sole food source for Monarch butterfly larvae[16]—and grasslands and adjacent wetlands in the US Prairie Pothole Region alone support over 50% of the North American breeding ducks historically surveyed by the US and Canadian wildlife services[17]. In addition, remaining grassland tracts that have never been plowed harbor some of the greatest concentrations of native plant species across the country[18,19]. Yet, despite increasing threats to these landscapes through cropland expansion[20,21], the habitat quality of lands that are typically converted remain largely unknown—hindering efforts to conserve affected biota or mitigate the impacts of conversion.

Collectively, the uncertainty surrounding the persistence, yields, and implications for wildlife of cropland expansion has encumbered evaluation of its merits and consequences. Given the geography and characteristics of new croplands and converted habitat in the US, we hypothesize that contemporary cropland expansion may provide diminishing production gains while engendering significant costs to wildlife. To assess this hypothesis, we map cropland expansion and abandonment throughout the United States between 2008 and 2016 and assess conversion locations as they relate to the anticipated yields of new and existing croplands and to the quality of wildlife habitat of public concern.

We begin by tracking field-level changes throughout the entire time series of nationwide USDA cropland maps[22]. Using these data, we improve upon previous work[9] and identify annual (rather than endpoint) changes over eight conversion years (e.g., 2008–09 = one conversion year) at a consistent 30-m spatial resolution. We then pair these results with modeled corn, soybean, and wheat yields to evaluate the representative productivity of new croplands in relation to that of pre-existing fields. Finally, we assess the impacts of conversion on the habitat of the

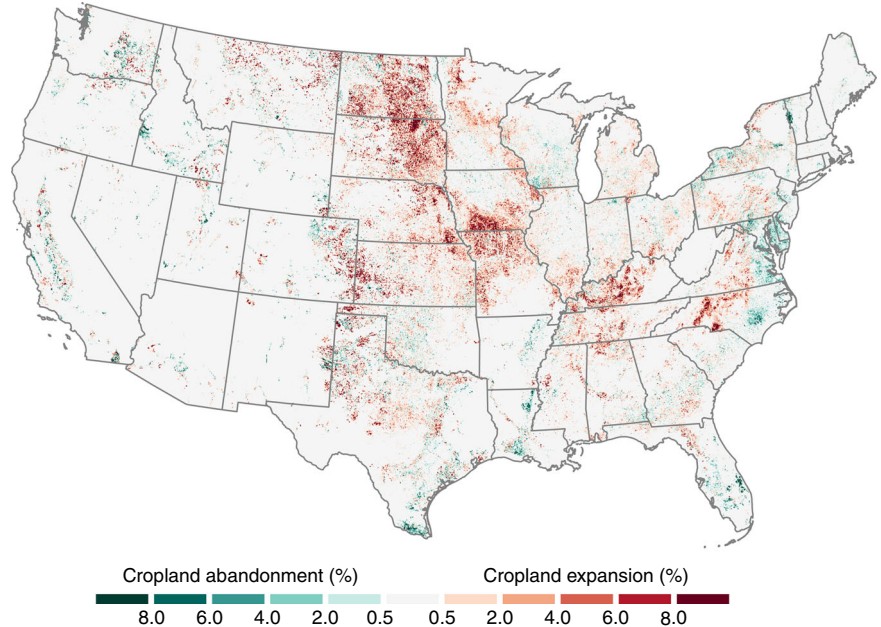

**Fig. 1 Net cropland conversion 2008–16.** Rates of net conversion calculated as gross cropland expansion minus gross cropland abandonment and displayed as a percentage of total land area within non-overlapping 3 km × 3 km blocks. Net conversion was most concentrated in the eastern halves of North and South Dakota, southern Iowa, and western portions of Kansas, Kentucky, and North Carolina.

Monarch butterfly (*Danaus plexippus*), a common pollinator; on migratory game species (nesting waterfowl); and on native plant communities (long-term grasslands). These specific taxa were selected for their familiarity to the public, representation of a broad range of wildlife types, and wide recognition as indicator species[23]. We find that cropland expansion has continued at a rate of over 1 million acres per year while generating below-average yields and supplanting high-quality habitat. Overall, our analyses provide insights into recent patterns of US land use change at higher spatial, thematic, and temporal resolution than any other source and foster an improved understanding of the agronomic and ecological trade-offs associated with changes in cropland extent.

## Results

**Spatial and temporal patterns of recent cropland changes**. In the 8 years following 2008, 10.09 million acres of land (1 acre = 0.40 hectares) were converted to crop production throughout the US (Fig. 1; Supplementary Table 1). Gross cropland expansion peaked in 2011, when 1.94 million additional acres were brought into cultivation before the rate stabilized at ~1 million acres per year for 2013–15 (Fig. 2). During the same 8-year period, only 3.52 million acres of cropland were abandoned or converted to noncrop uses, with a maximum annual gross rate of 1.06 million acres in 2010. The rates of net cropland conversion, or gross expansion minus gross abandonment, ranged from 0.38 to 1.33 million acres, which indicates a continued and persistent increase in US cropland area.

Consistent with earlier findings[9,24], the Prairie Pothole Region (PPR) of North and South Dakota, the Dissected Till Plains of Iowa and Missouri, and the High Plains portions of Kansas, Oklahoma, and Texas remained pre-eminent hotspots of

expansion. However, locations extending along the Canadian border of the Northern Great Plains and the Interior Low Plateau of Kentucky and Tennessee have more recently emerged as additional nuclei of activity. Rates of cropland abandonment, on the other hand, have been greatest along the Mid-Atlantic Coast, the Gulf Coast, and parts of the Pacific Northwest (Supplementary Figs. 1-3).

Grasslands, including those used for pasture and hay, constituted 88% of the land converted to crop production across the US. Regional patterns in the loss of other natural land-cover types included the clearing of shrublands in western states, the cultivation of wetlands across the PPR, and the conversion of forest in the southeastern US (Supplementary Fig. 4). Overall, the highest rates of loss of natural landcover relative to its remaining area occurred in swaths of the western Corn Belt and western Plains, where rates of existing cultivation and cropland expansion were both high (Supplementary Fig. 5).

**Yields of new croplands relative to existing crop extent**. Corn was the predominant crop planted on newly cultivated land; it was most common in all years except 2014–15, when soybeans were more prevalent (Supplementary Figs. 6-7). Together with wheat, these three crops were the first plantings on over 78% of all new croplands nationwide. Compared with the national average of existing croplands, the representative yields of new croplands were 10.9% lower for those planted to corn (SD$_{spatial}$ [standard deviation of spatial variation] = 13.8%), 8.4% lower for soybeans (SD$_{spatial}$ = 14.9%), and 1.3% higher for wheat (SD$_{spatial}$ = 21.1%) (Supplementary Table 2). Across the US, yields for corn, soybeans, and wheat were less than their corresponding national averages on 78%, 69%, and 59% of new croplands, respectively (area-weighted average of 69.5%).

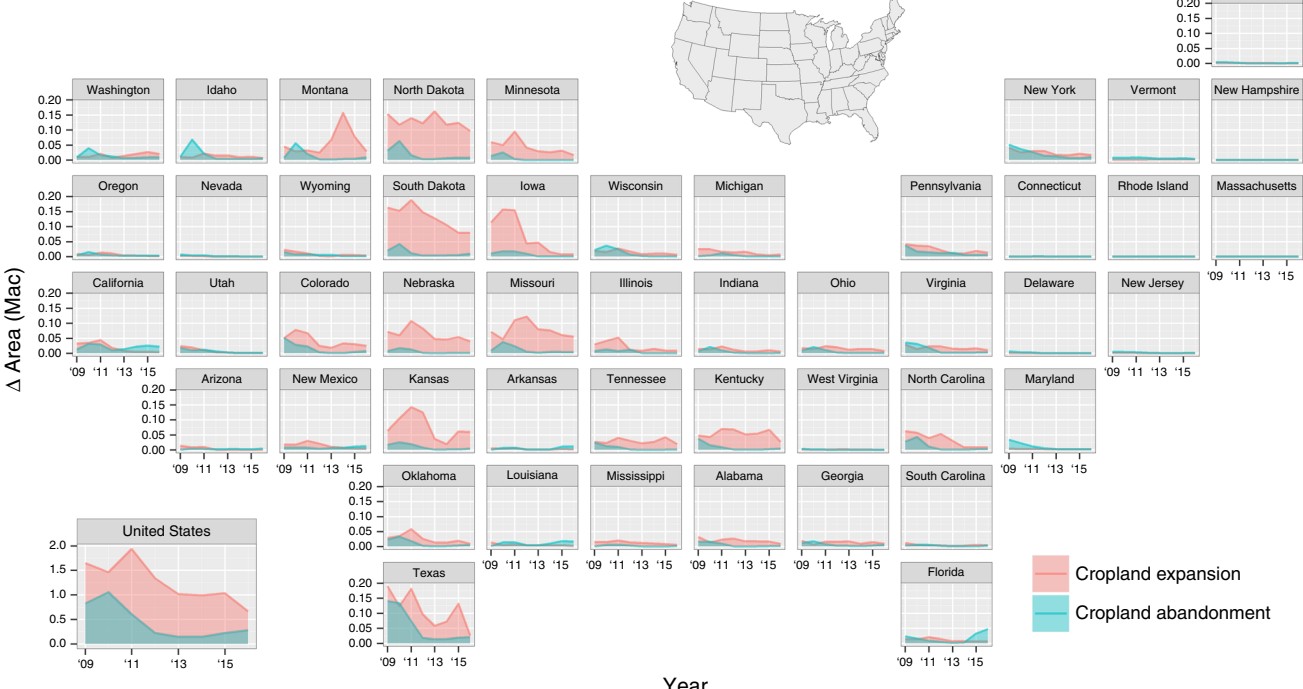

**Fig. 2 Annual cropland expansion and abandonment by state, 2008–16.** The values for each year reflect the gross areas of cropland expansion and abandonment that occurred between that year's growing season and that of the previous year—i.e., conversions for 2008–09 are reported as '09. Iowa, Kansas, South Dakota, and Texas experienced some of the greatest transformations to cropland, with their rates of expansion cresting in earlier years and falling over time. Minnesota, Missouri, and Nebraska exhibited similar temporal trends but at lower magnitudes. High rates of expansion in North Dakota and, to a lesser extent, Kentucky, were persistent across the full study period. Montana was the only state with greater conversion to cropland after 2012 than before.

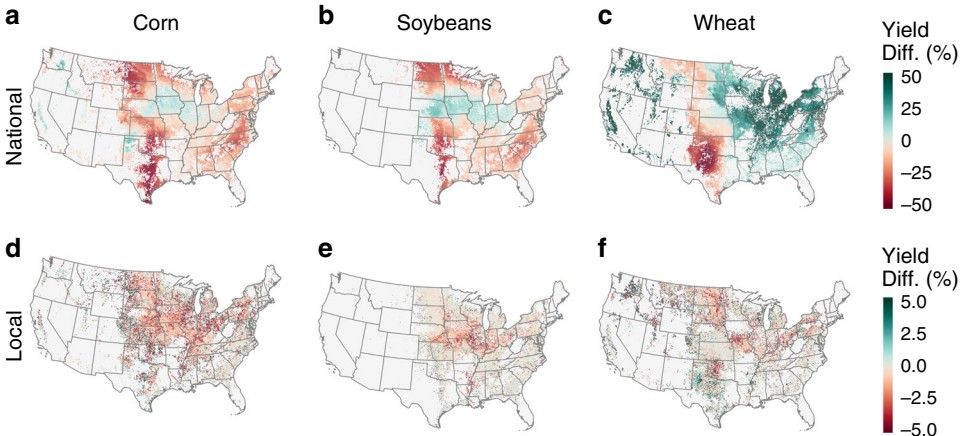

**Fig. 3 Yield differentials of new croplands compared to existing croplands.** Yield differential values represent the yields of new croplands relative to the yields of existing croplands nationwide (**a**–**c**) or within immediate 10 km x 10 km neighborhoods (**d**–**f**). Yields of new croplands planted to corn and soybeans were typically lower than the national average (**a**, **b**) and the nearby local average (**d**, **e**) of existing croplands for each crop. Yields of new croplands planted to wheat were generally higher than the corresponding national average (**c**) but lower than nearby existing croplands (**f**).

Expected corn and soybean yields were lowest in the expansion hotpots of the northern and southern Plains, where new croplands frequently achieved just half of the nationwide average yields of existing croplands (Fig. 3a–c). Yields of new croplands planted to wheat were projected to be lowest in Texas and along the western edge of the Plains but higher than average throughout the Midwest and eastern US, where wheat is less frequently grown. For all three crops, yields on new croplands in regions that were already highly cultivated fared better in relation to nationwide means (Supplementary Fig. 8a–c and Supplementary Table 3).

At local scales, new croplands have also encroached upon poorer quality land, though the differences relative to current croplands were generally smaller than the national-scale differences. Compared with yields of existing croplands within their immediate 10 km × 10 km neighborhood, yields of new croplands were on average 1.1% lower for corn (SD$_{spatial}$ = 1.8%), 0.6% lower for soybeans (SD$_{spatial}$ = 1.1%), and 0.7% lower for wheat (SD$_{spatial}$ = 2.9%) (Supplementary Table 4). These local yield differentials varied widely across the US, though the largest disparities generally occurred in highly cultivated areas such as the eastern Corn Belt, where new fields commonly yielded >5% less than nearby existing fields (Fig. 3d–f). This pattern was also universal—areas with greater cultivation and less remaining natural land had significantly larger local yield deficits (Supplementary Fig. 8d–f and Supplementary Table 3).

Our modeled yield results are consistent with the rated suitability and agronomic characteristics of new and existing croplands. Across the US, new fields were generally less suited to cultivation and presented greater limitations according to the USDA's land capability classification system (Supplementary Fig. 9 and Supplementary Table 5). Likely contributing to this broad measure of suitability, new croplands were characterized by several biophysical disadvantages. We found, for example, that the average slope gradient of new croplands was 3.35% (SD$_{spatial}$ = 3.45), or ~1.7 times greater than that of existing croplands (mean = 2.00%, SD$_{spatial}$ = 2.60) (Supplementary Fig. 10). Crop production is also shifting into more arid climates; during our study period new croplands experienced on average a 3.3% higher climate water deficit, calculated as the amount of evaporative demand that is not met by available soil water[25]. New croplands were also less frequently planted on hydric soils (8.10% vs 19.19%), or at those locations for which the topsoil is water-saturated for at least part of the year, when compared with existing croplands (Supplementary Fig. 11).

**Wildlife habitat impacts**. We estimate that ~220 million (SE ± 189) common milkweed stems were lost due to conversion of grasslands, wetlands, and shrublands to corn and soybean production across the Midwest during our study period. This loss represents 8.5% of the estimated regional total in 2008. The largest reductions occurred in the Dakotas, Iowa, and Missouri, due to a confluence of high rates of conversion with high proportions of conversion from lands enrolled in the Conservation Reserve Program (Fig. 4a; Supplementary Figs. 12-13; Supplementary Note 1), which harbor greater densities of milkweed stems per acre than pasture, hayland, and other noncrop uses[16,26,27]. On average, natural land converted to cropland contained an estimated 53.7 (±46.0) stems per acre prior to conversion, which is 3.4 times greater than the 15.6 (±10.4) stems per acre on all existing natural lands in the region (Fig. 4b).

In the Prairie Pothole Region (PPR), grasslands and wetlands are also particularly valuable for waterfowl reproduction[28,29]. We assessed recent cropland expansion in relation to duck breeding pair accessibility across the PPR and found that areas estimated to provide 138,000 nesting opportunities (2.8% of the regional total) were recently converted to crop production (Fig. 5). On average, converted habitat was in locations accessible to an estimated 42.7 breeding pairs per square mile (SD$_{spatial}$ = 30.6), which is nearly twice as high as the average for existing croplands (22.9; SD$_{spatial}$ = 24.7) and 37% greater than other habitat that was not converted (31.2; SD$_{spatial}$ = 30.2).

Finally, we observed substantial conversion of long-term habitat, here defined as locations that would not have been cultivated for cropland or pasture for at least a quarter century. These areas often contain disproportionately high numbers of native plant species and undisturbed sod. During the study period, 2.8 million acres of new cropland (28%) came from these longstanding habitat sites, of which 2.3 million acres (81%) were unimproved grasslands (Fig. 6). Relative to all converted land, 26% of grasslands, 29% of wetlands, 44% of forest, and 52% of shrublands that were converted met this criterion for longevity.

## Discussion

Our analyses confirm the continued and widespread expansion of croplands across the US and reveal the diminished production benefits and disproportionate costs to wildlife associated with this activity.

Our results are similar in direction and spatial pattern to other, coarser estimates of cropland expansion and abandonment,

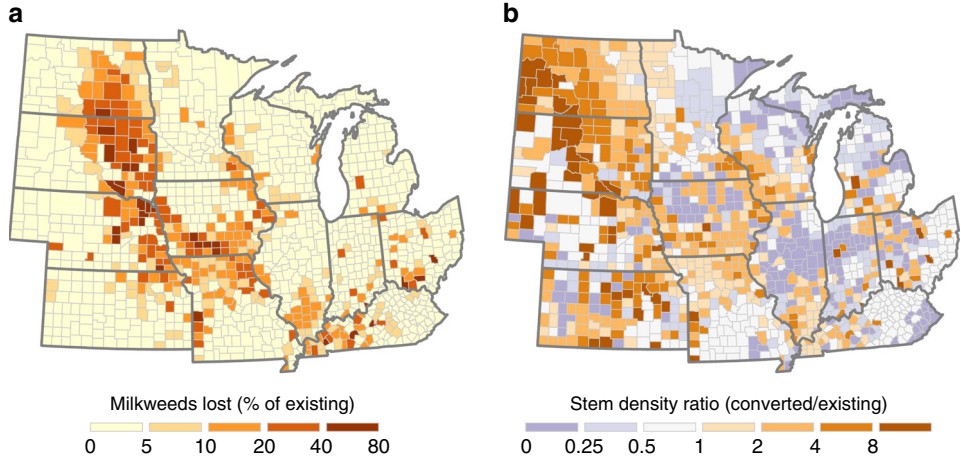

**Fig. 4 Monarch habitat loss due to conversion of land to corn and soy production in the Midwest, 2008–16.** The maps represent the number of milkweed stems lost as a proportion of those on all grasslands, shrublands, and wetlands in 2008 (**a**), and the density of milkweed stems on land subject to conversion relative to the density on existing lands in 2008 (**b**). Large losses of milkweed occurred in the region stretching from eastern North Dakota to northern Missouri—locations with high rates of cropland expansion and conversion from Conservation Reserve Program (CRP) lands, which harbor high densities of milkweed stems. Across the Midwest region, the average density of milkweed stems on land prior to conversion was over three times greater than that on average existing land. Milkweed stems provide the sole food source for Monarch butterfly larvae.

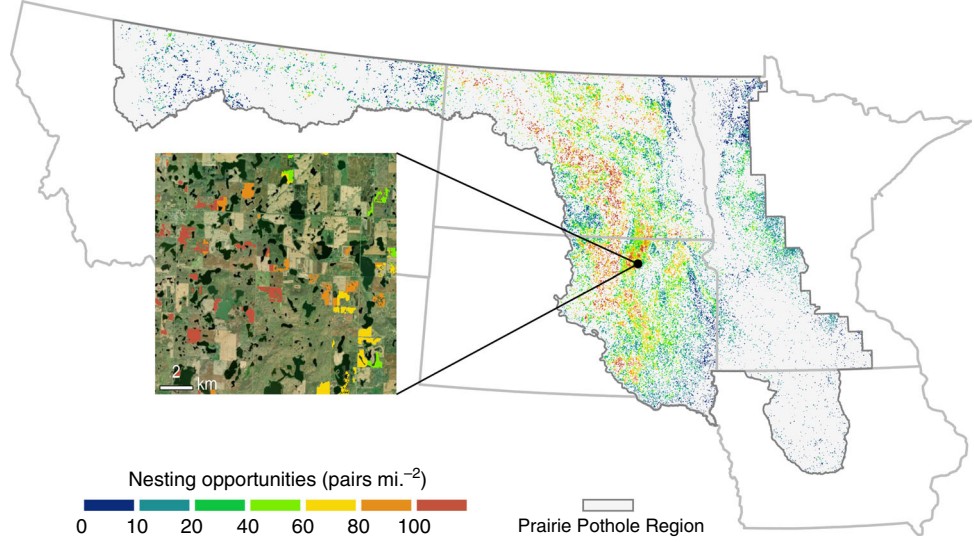

**Fig. 5 Estimated duck breeding pair accessibility of land converted to crop production in the Prairie Pothole Region of the northern US, 2008–16.** Values are displayed for all areas of noncropland converted to cropland within the PPR modeling extent (gray background). Nesting opportunities reflect the estimated number of duck pairs within a one square mile range which could access available habitat. Conversion was concentrated around the Missouri Coteau region of eastern North and South Dakota, where the accessibility of nests by breeding pairs is particularly high. The inset map depicts an example of the field-level results in the region, with converted fields colored according to their estimated accessibility to nesting pairs.

including the USDA Census of Agriculture (CoA)[30], the USDA National Resources Inventory (NRI)[31], and the USGS National Land Cover Database (NLCD)[32] (Fig. 7; Supplementary Figs. 14–15; Supplementary Table 6). Together, these major assessments utilize three distinct approaches: farmer surveys, high-resolution photo interpretation with field validation, and satellite remote sensing. Despite the independence of their data sources, these disparate analyses demonstrate a clear consensus of extensive cropland expansion over the past decade in the US.

The magnitude of our estimates of gross conversion are generally more conservative than others, due in part to our prioritization of improved map confidence in converted areas over complete capture of all conversion (see Supplementary Note 2; Supplementary Table 7). The uncertain nature of the underlying remote sensing products may further contribute to dissimilarities.

For example, it can be challenging to analyze recent abandonment from satellite data due to confusion among active fallow and noncropland classes, and the general difficulty of differentiating short-term idling of land from longer-term abandonment[33–35]. Despite these limitations, field-level expected accuracies for all cropland expansion and abandonment ranged from 71.0 to 86.9% for both the user's and producer's perspectives (Supplementary Table 8) and were consistent with targeted standards for measuring change[36,37].

We found that croplands are moving onto lower-quality land in less-suitable regions—a dual setback to production gains from cropland expansion. The national yield differentials that we observed largely mirror productivity gradients that reflect broadscale patterns in climate and landscape suitability[38–40]. The prevailingly negative differentials—or yield deficits—of new

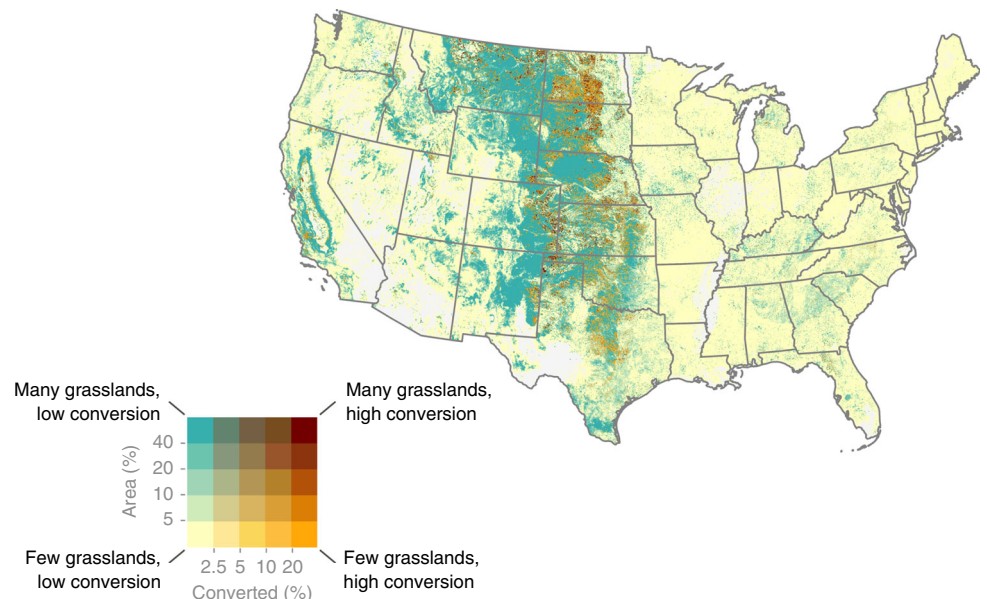

**Fig. 6 Long-term grasslands converted to crop production 2008–16.** Long-term grasslands are those that have not been planted or cultivated for at least 25 years and are most likely to contain native plant species and undisturbed sod. Values displayed reflect the percentage of the landscape that is occupied by long-term grasslands (y-axis) and the percentage of those grasslands that were converted (x-axis). Conversion (orange color) was highest along a north-south transect of the Great Plains, which represents a westward frontier of cropland expansion into remaining long-term grasslands.

croplands therefore suggest expansion is occurring in regions less conducive to agricultural production. In contrast, local yield differentials capture the finer scale productivity differences among new and existing croplands within a single climate and landscape and thus reflect variation mostly in soil and topography. The prevalence of local yield deficits confirms that individual farmers looking to expand their operations are generally confined to cultivating increasingly marginal land, though widespread variation existed from location to location. These field-level findings corroborate and extend earlier estimates showing that yields of land moving into and out of the CRP specifically are lower than average for both their agricultural district[41] and nationwide[42].

We also detected a broader relationship between the magnitude of yield deficits and the remaining quantity of arable land. In locations with limited land left to cultivate, local marginality (but not national-scale marginality) is more pronounced—a phenomenon that reflects high competition for land and is consistent with theories of scarcity and agricultural rent[43]. The loss of relatively rare patches of uncultivated lands in these locations may also be notably detrimental to wildlife, as these areas often represent important and nonredundant migratory corridors and refugia[44,45]. Conversely, regions with substantial tracts of remaining natural landcover exhibit less local marginality and competition for land, but yields are significantly lower than their respective nationwide averages. The effects of cropland expansion on natural habitat in these locations may thus be more diffuse but the associated production returns are relatively diminished.

Yield deficits result from biophysical conditions that may have additional implications for farmers and the surrounding environment. New croplands had steeper slopes than existing croplands, which may limit certain cultivation practices[46], pose safety hazards to farmers[47], and increase the risk and magnitude of soil erosion and its ensuing effects on water quality[31]. New croplands also occupied more arid regions that had larger climate water deficits and are often associated with either greater susceptibility to drought or increased need for irrigation[48,49]. Surprisingly, expanded production was less likely to occur on hydric soils,

which indicates lower conversion rates of these wetland-capable locations at present than during the historical establishment of croplands in the US.

Across diverse ecosystems and wildlife types, the loss of habitat associated with cropland expansion comes at considerable cost. Monarch butterflies, though themselves not a leading pollinator, are key indicators of insect biodiversity and a flagship species for pollinator conservation[23,50,51]. We show that the contribution of land conversion to the annual loss of milkweed stems is 11- to 14-times larger than previously reported[52]. This puts the impacts of recent land use change on the same order of magnitude as the widespread extirpation of milkweed caused by GMO crop development and associated pesticide use over the last two decades—historically considered the pre-eminent threat to Monarchs[53,54]. Our results embody significant uncertainty, however, and recent field surveys suggest that milkweed concentrations on many types of grasslands prone to conversion may be even greater than estimated here[27] (see Supplementary Note 1).

The continued loss of milkweed from ongoing land conversion may also pose a threat to future recovery efforts. The current habitat restoration target—addition of 1.3 billion more stems in order to return Monarch populations to pre-1990s levels—already requires an "all hands on deck" approach to maximize milkweed in every land sector[16]. If cropland expansion continues along the trends reported here, there may be scant opportunity to accommodate and restore the nearly 30 million additional stems lost to land conversion each year in the Midwest landscape.

Conversion to cropland similarly embodies a loss of habitat for waterfowl and other game species. We found that grasslands and wetlands in locations subject to conversion had greater nesting accessibility for ducks than either croplands or other natural areas without conversion. These results indicate that disproportionately large reductions to breeding habitat and reproductive capacity could be expected with continued cropland expansion. Moreover, this finding substantiates conservation strategies that aim to protect the natural lands at highest risk of conversion in order to maximize wildlife-supporting benefits[29,55,56].

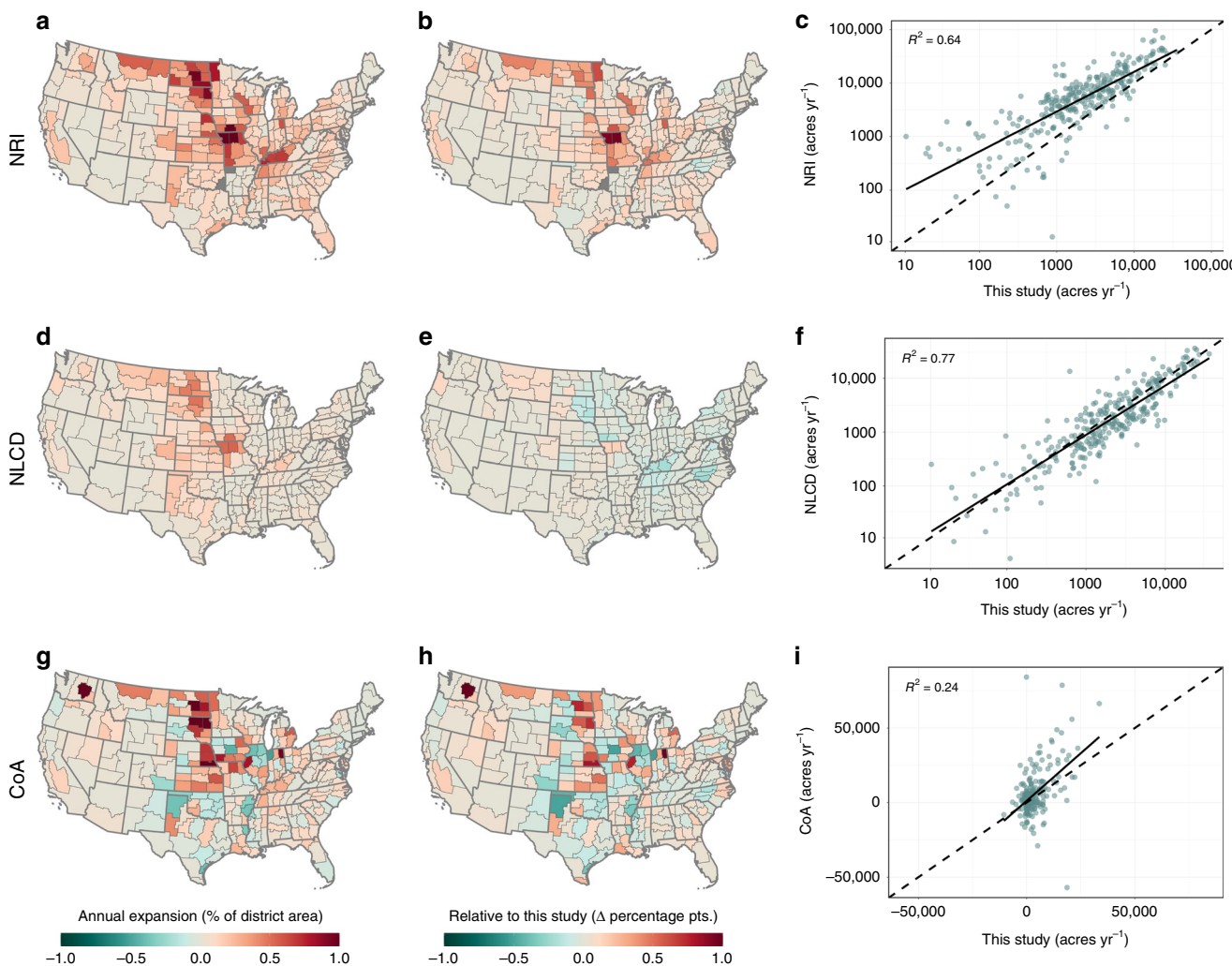

**Fig. 7 Comparison to other recent cropland expansion estimates.** Average annual rates of cropland expansion derived from the National Resources Inventory (NRI) 2007–15 (**a–c**), the National Land Cover Database (NLCD) 2008–16 (**d–f**), and the Census of Agriculture (CoA) 2007–17 (**g–i**). Maps in the first column (**a**, **d**, **g**) depict the annual rate at which croplands expanded within each agricultural district, calculated as a percentage of the total district area. Maps in the second column (**b**, **e**, **h**) show how these annual rates differ in absolute terms (Δ percentage points) from the comparable estimates derived in this study. Scatterplots in the third column (**c**, **f**, **i**) illustrate how absolute annual rates of expansion within each district (acres yr⁻¹) from each data source compare to those in this study. The $R^2$ value for each linear regression is listed on the corresponding plot, with full regression statistics and state-level results reported in Supplementary Table 6 and Supplementary Fig. 14. Note that NRI and NLCD estimates and comparisons to this study are based on rates of gross cropland expansion; CoA estimates and comparisons based on rates of net cropland change. All three comparison datasets—based on unique sources—corroborate the general trend and spatial patterns observed in this study of widespread cropland expansion throughout the US over approximately the past decade.

The subset of conversion which occurs on long-term habitat is likely to lead to particularly high losses in native plant diversity, wildlife provisioning, and other ecosystem services[18,19]. The large area of long-term grassland conversion we identified provides strong evidence of widespread clearing of previously undisturbed lands, especially when considered alongside other reports of confirmed native prairie conversion throughout the US[20,21,57,58]. This ecologically significant loss of intact habitat indicates that if remaining virgin land is to be preserved, stronger protections will be required such as those afforded through federal policy, conservation incentives, and/or supply chain interventions[59].

Collectively, our analyses of representative pollinator, waterfowl, and native plant habitats show that cropland expansion is infringing upon high-quality natural land in such a way as to disproportionately affect the wildlife that depends on it. Given these indications, future analyses might aim to further examine the impacts of expansion on a wider range of taxa[60,61]—especially

endemic species—and how they manifest throughout the population, community, and ecosystem levels. Such biodiversity impacts should also be compared with other trade-offs, such as effects on climate, water use, and water quality.

Overall, our results have noteworthy implications for the global agronomic, conservation, and policy communities. For example, projections of future crop supply[62,63] and estimates of crop area-yield elasticities[64–66] could be refined using the diminished yield returns that we observed for additional cropland areas. Similarly, evaluations of the trade-offs inherent in expanded crop production should consider the higher wildlife costs and lower yields of converted lands, particularly when assessing impacts on a per-unit basis[2,7,67,68]. Such consideration may likewise clarify the ongoing debate over the relative merits of cropland expansion versus intensification as a means of increasing total production[69].

The tacit side effects of cropland expansion reinforce the notion that continued agricultural extensification under current

practices cannot sustainably shoulder the burden of scaling society's food, fiber, and fuel production systems without compromising the planet's supply of ecosystem services[70,71]. This recognition escalates the urgency of alternative solutions[72,73]. Other activities, like minimizing food waste[74,75], reducing animal-based diets[4,76], and closing yield gaps[77,78] will need to play more prominent roles in meeting future agricultural demand while maintaining ecological integrity. Likewise, practices that improve efficiency[79,80] and reduce the impacts of current cropland management—such as the use of cover crops[81,82] vegetation strips[83,84], diversification of crop rotations[85], and restorative agriculture[86–88] increase in importance in a world that is more reliant on the existing extent of cropland area. Whereas further cropland expansion is likely to exacerbate the agronomic risks, yield deficits, and wildlife impacts of increased production, the alternative of slowing or halting this ongoing land conversion offers a vital opportunity to help address the growing challenges facing agricultural production systems.

## Methods

**Study design**. Our study was designed and conducted to answer the following three research questions: (1) What are the annual spatiotemporal patterns of recent cropland expansion, particularly following its resurgence at the end of the 2000s decade? (2) How do the yields of new croplands compare to those of existing fields, and how do these differences vary across space and scales? (3) What are the absolute and relative impacts of recent cropland expansion on wildlife habitat that are of public concern? To address these questions, we paired analyses of land use change with those of crop yields and habitat quality to assess the trade-offs of recent land conversion on both agricultural and natural ecosystems. Our time frame of 2008–16 corresponds with the availability of crop-specific land use data[22] and encompasses a variety of market and environmental conditions, including periods of both high and low crop prices, as well as drought, normal, and wet years, and thereby provides insights into the more persistent characteristics of cropland expansion.

**Conversion detection**. Land use changes were identified from time-series analysis of the USDA Cropland Data Layer (CDL) and National Land Cover Dataset (NLCD) following the general approach of Lark et al.[9]. We implemented several new methodological advances: these included tracking of changes on an annual basis, improved spatial resolution from 56 to 30 m, and crop-specific spatio-temporal filtering[89,90]. The conversion detection algorithm is summarized below, with full method and dataset details available in the Supplementary information (see Supplementary Methods). In general, the processing consisted of compiling data, refining them via class-specific and nonspecific treatments, and assessing coupled attribute data to determine specific years and classes of conversion.

First, we consolidated all classes of the CDL into two categories—cropland and noncropland—for each year of CDL data 2008–17 in order to reduce the amount of data upon compilation and facilitate detection of broad agricultural land use changes. Cropland was broadly defined as any area planted to cultivated row, closely grown, or horticultural crops and included cultivated fallow and alfalfa (Supplementary Table 9). The 2001, 2006, and 2011 NLCD products[91] were similarly reclassified into binary cropland or noncropland designations based on the cultivated crop category (class 82). We then combined all data into a trajectory layer where each pixel value represented a unique temporal pattern of crop or noncropland use. To these trajectories we applied a majority spatial filter with parameters of eight neighbors and a replacement threshold of one-half of the spatially contiguous pixels. This filtering enhanced classification of mixed-use pixels located at the edge of feature boundaries[92] and because it was applied to the multiyear composite of images also aided removal of pattern anomalies across both space and time.

After spatiotemporal filtering, we reclassified the trajectories into five general land transition classes: (i) stable noncropland; (ii) stable cropland; (iii) conversion to cropland; (iv) conversion to noncropland; (v) intermittent cropland following a set of reclassification rules (see Supplementary Methods). In general, a conversion between cropland and noncropland was defined as any pixel classified as cropland (or noncropland) in both the two preceding CDL years and the two previous NLCD products, and then subsequently classified as the opposite class in the two succeeding CDL years. Thus, areas converted to cropland must have been considered noncropland for at least 6–10 years prior to their conversion (based on a composite of CDL and NLCD data) and must have remained cropped for the 2 years immediately following conversion. Furthermore, only those trajectories that underwent a single transition to or from cropland were considered conversions. Locations which were cropped for at least 2 of the 10 years of CDL input data but switched between cropland and noncropland more than once were classified as intermittent (rotational) cropland.

After classifying the data into these five broad transition categories, we applied a minimum mapping unit of five acres to match the detection capability of the input data with the resolution of features on the landscape[89]. Contiguous patches less than five acres in size were removed, and the resulting void pixels replaced with the transition class of the nearest remaining patch, based on Tobler's first law of geography: near things are more similar than distant ones[93,94]. Note that we implemented the minimum mapping unit on the five aggregated land transition classes—which each represent a composite of input data and conversion years—rather than on a single year of land use or conversion data. This sequencing is an important facet for detecting change because fields that have recently been converted often have greater uncertainty (as indicated by the CDL confidence layer) and heterogeneity in their classifications[89,90]. For example, newly converted fields are occasionally classified in salt-and-pepper patterns with part of the field labeled as the new, correct landcover class in the first year after a conversion and part of the field remaining labeled as the previous, incorrect landcover class until the subsequent year. Thus, if a minimum mapping unit or other spatial filter is applied to individual years of input or conversion data, these mixed-year conversion areas may be missed or removed.

While our data are ultimately dependent upon the classification accuracy of the CDL, we also implemented a number of class-specific refinements to certain crops and landcovers to correct for known CDL errors and uncertainties in measuring change. For example, we excluded from our assessment any conversion among fallow/idle cropland or alfalfa (both considered cropland) and non-alfalfa hay or grassland/pasture (both considered noncropland) because these four classes are frequently confused[90]. Similarly, we excluded any potential conversion of developed land to cropland due to the unlikelihood of these transitions. Additional special treatments were utilized for perennial tree crops and rice (see Supplementary Methods).

After the locations of conversion were identified, we characterized the year of conversion and the specific landcover preceding and following a conversion for all areas of change. These layers were processed in a manner consistent with the broad transition categories described above to maintain consistency throughout the analysis. This general approach, that is, to first assess conversion using broad cropland versus noncropland categories and then to re-identify specific landcovers, leveraged the high accuracy (>98% for all years) of the CDL in delineating crop from noncrop areas while it also retained the thematic richness of the original classification[89,90].

**Yield modeling**. We used crop-specific random forest (RF) models to predict representative grain yields of corn, soybeans, and wheat on newly converted croplands using a suite of biophysical covariate grids. Unlike traditional yield modeling methods that often require a lengthy observational record (e.g., remote sensing or process-based approaches), our method allows assessment of likely yields of new fields which may lack such a record. Moreover, our objective was not to estimate yields year-to-year with unprecedented accuracy or in-season time-liness, as is often the goal of traditional approaches, but rather to broadly assess the production potential of new croplands in relation to those that already exist. Our models, therefore, do not account for dynamic factors like stochastic variation in weather, anomalous management, or genetic improvements. Rather, their predictions are intended to represent average expected yields within the period of our training data (2008–17) and are a function of the local biophysical setting and the management practices implicitly associated with those conditions. For the sake of methodologically consistent comparison, we applied our models to both newly converted and stable pre-existing cropland classes and report only relative (%) differences to ensure proper interpretation of the model's predictions.

Random forest is a nonparametric, data-driven method that generates predictions based on an ensemble of bootstrapped classification or regression trees[95] and has been successfully used by others for yield predictions[96] akin to ours. We developed three RF models (one each for corn, soybeans, and wheat) using training datasets we collated from annual county-level yield averages for each of the three crops and the corresponding means of biophysical covariates within each crop's planted extent in a given year and county. We used 10 years (2008–17) of county-level crop yield averages from the USDA's Agricultural Resource Management Surveys[97]. In these years we could precisely determine the location of each crop in each county using the CDL. We retained yield data for all US counties for which they were available and did not differentiate data according to irrigation status nor, in the case of wheat yields, among varieties (e.g., spring, winter, or durum).

Within each county we determined the biophysical characteristics associated with each of the yield observations by tabulating the mean values of gridded covariates (Supplementary Table 10) within the CDL-determined planted area of each crop and year. Covariates included the mean, sum, minimum, and maximum values of multiyear means (2008–17) of monthly gridded climate and water balance metrics from the TerraClimate database[25]; the National Commodity Crop Productivity Index (NCCPI) for (i) corn/soybeans and (ii) small grains from the gSSURGO soils database[98]; and elevation, slope, and aspect grids derived from the National Elevation Database (NED)[99] in Google Earth Engine (GEE)[100]. TerraClimate grids had a native spatial resolution of 2.5 arcmin and were resampled to 30 m prior to tabulation using the bilinear method in GEE to match the resolution of the CDL and the NCCPI grids. Grids derived from the NED had a

1/3-arcsec resolution and were aggregated in GEE to a 30-m resolution. Tabulated covariate statistics were then joined to the corresponding yield data to complete the training set.

Once the three crop-specific training datasets were amassed, they were used to generate separate RF models for each crop. The three training datasets were each comprised of roughly 13,000 data records, of which we randomly selected and withheld 30% for model validation (Supplementary Fig. 16). Beyond a training dataset, RF models also require three additional parameters: (i) the number of regression trees generated for the ensemble (ntree), (ii) the number of variables considered at each split of those trees (mtry), and (iii) the minimum number of points considered in each node of those trees (node size). We used the randomForest package in R[101] to select parameter values that minimized the mean square error of predictions when compared with the validation set. For all three models, we found the optimal parameter set to be 250 trees, with 21 variables considered at each split, and a minimum node size of five points. These parameters and each of the training datasets were then supplied to random forest regression classifiers in GEE where they were implemented spatially using the covariate grids to map representative yields for all newly converted and existing stable cropland parcels. Variable importance plots for each crop model are presented in Supplementary Fig. 17.

We then used these yield estimates to calculate the national yield differential (diff_nat) of each grid cell as the relative deviance of the cell's expected yield of a given crop from that of the national average yield of the respective crop (Eq. (1)):

$$\text{diff}_{\text{nat}} = \frac{y_{\text{gc}} - y_{\text{nat}}}{y_{\text{nat}}} \qquad (1)$$

where $y_{\text{gc}}$ is the expected crop-specific yield of that grid cell for new croplands and $y_{\text{nat}}$ is the national average yield of that crop for existing croplands according to the model. The $y_{\text{nat}}$ of each crop was calculated as the frequency-weighted average of predictions within the extent of the stable cropland class (defined above) in which the focal crop (corn, soybeans, or wheat) was grown at any time during the study period (2008–16).

A local yield differential (diff_loc) was similarly calculated (Eq. (2)) by comparing a local 30-m resolution grid cell's yield for new croplands ($y_{\text{gc}}$) to that of the existing croplands within the grid cell's encompassing 10 km × 10 km neighborhood ($y_{\text{neigh}}$).

$$\text{diff}_{\text{loc}} = \frac{y_{\text{gc}} - y_{\text{neigh}}}{y_{\text{neigh}}} \qquad (2)$$

We subsequently calculated the national average of the local differential as the area-weighted mean of all 10 km grid cells, with weights determined by the extent of new croplands within each 10 km cell.

**Identifying converted land characteristics.** We used slope information from the National Elevation Dataset[99] along with data on land capability class (LCC) and hydric soil status from the gSSURGO database[98] to assess the relative biophysical characteristics of new and existing croplands. Slope percentages represent the grade of converted land calculated as the amount of elevation change per unit of horizontal distance (i.e., rise over run). The LCC system estimates the capability of land to support nonirrigated agriculture according to increasing levels of restrictions and limitations to cultivation[46]. Hydric soils are those that are formed under conditions of water saturation, flooding, or ponding long enough during the growing season to develop anaerobic conditions in their upper part[102]. Mean annual climate water deficit (2008–17) was calculated using monthly grids from Terraclimate[25], wherein climate water deficit is defined as the difference between a reference evapotranspiration (in this case, potential evapotranspiration) calculated using the Penman–Monteith approach and actual evapotranspiration.

**Assessing habitat impacts.** We estimated the losses and associated uncertainty of milkweed stem numbers following the method of Pleasants (2016)[52]. In addition, we also specifically considered the areas of converted grasslands that were previously enrolled in the Conservation Reserve Program (CRP)[52]. This extra step, which delineates converted grassland types, enables better approximation of the densities of milkweed stems on converted land and their variation across counties (see Supplementary Note 1). We estimated duck breeding pair nesting accessibility using US Fish and Wildlife Service thunderstorm maps ca. 2012; these reflect the spatial variation in habitat quality across multiple years of environmental conditions near the median of our study period[28,103]. To estimate breeding pair accessibility for each land use type, we used the midpoint value for each category range provided (e.g., a value of 70 pairs/sq. mi. for the range of 60–80 pairs/sq. mi.). For the highest category, >100 pairs/sq. mi., we assumed a density of 110 pairs. Total accessibility of nesting opportunities was then averaged across the full PPR for each land transition category. To estimate long-term grasslands, we used data from the 1992, 2001, 2006, and 2011 NLCD[91] to identify land that had not previously been used for crop production or planted pasture/hay. Locations which had never been classified as cropland (class 82) or pasture/hay (class 81) in any year of the NLCD were considered to be long-term and unimproved and were used to approximate the areas with potential to contain native prairie as well as the subset of conversion occurring on those lands.

**Accuracy estimation and comparison of results.** To estimate the accuracy of a conversion between noncropland and cropland, we calculated the likelihood a pixel was correctly identified broadly as cropland or noncropland for each state and specific landcover class (see Supplementary Methods)[90]. These superclass accuracies were then used to estimate the likelihood that a conversion was correct by multiplying the state- and class-specific superclass accuracy of each converted pixel for the specific year and class preceding and following conversion (Eq. (3)). This approach provides a thematically and temporally explicit estimate of the expected accuracy for each land use change identified.

$$\text{Expected Accuracy} = \text{SA}_{\text{yoc}} * \text{SA}_{(\text{yoc}-1)} \qquad (3)$$

where SA is the superclass accuracy and yoc is the year of conversion.

We also compared our results with those from the 2015 USDA National Resources Inventory[31], the 2017 USDA Census of Agriculture[30], and the 2016 USGS National Land Cover Database[32] (Supplementary Table 7). Mapped NRI values represent changes from noncropland to cropland, where cropland includes both cultivated and non-cultivated crops (i.e., horticultural and hay crops). From the CoA, we mapped the change in actively cultivated cropland by including all planted, failed, and fallow cropland and excluding idle cropland and cropland-pasture. These accountings most closely conform to the definition of cropland used in this study, and also follow the approach of recent reviews of land use products that aim to appropriately compare estimates across products[2]. For the NLCD, we mapped all conversion to cultivated crops (class 82) from other classes. Note also that the version to which we compare (NLCD 2016)[32,104] reflects a complete remapping of the NLCD series, and is independent of the older generation of data incorporated into our trajectory-based land change analysis[91]. We aggregated pixel-level results for the NLCD and our data and county-level results from the CoA to the agricultural district level in order to facilitate comparison with the NRI, for which agricultural districts are the finest resolution that offer complete representation of the data[31]. For thoroughness, we also aggregated and compared all datasets at the state level (Supplementary Fig. 14), as well as mapped the available partial datasets at the county level for visual comparison only (Supplementary Fig. 15). Note that we did not compare data reported by the United Nation's FAOSTAT database as those metrics are based upon USDA data, including the CoA estimates of total cropland. In addition, we excluded comparison to USDA NASS Survey planted area data[97] due to multiple confounding issues that preclude its use for estimating cropland change, e.g., incomplete spatial coverage, annual fluctuations in planting conditions, and its limited reporting of net rather than gross changes, planted area instead of active cropland, and particular principle crops rather than all crops (see Supplementary Note 2).

**Reporting summary.** Further information on research design is available in the Nature Research Reporting Summary linked to this article.

## Data availability

The land conversion datasets and modeled yield estimates generated in this study are viewable online at http://www.ag-atlas.org and have been permanently archived with Zenodo (https://doi.org/10.5281/zenodo.3905242).

## Code availability

The code developed for and used in this study can be found on GitHub (https://github.com/gibbs-lab-us/usxp_08_16) and has been permanently archived with Zenodo (https://doi.org/10.5281/zenodo.3905556).

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

## Acknowledgements

We thank Beichen Tian and Ian Schelly for their help in data analysis as well as George Allez and Alyssa Braun for editing the manuscript. This material is based upon work supported in part by grants from the National Wildlife Federation and the Great Lakes Bioenergy Research Center, U.S. Department of Energy, Office of Science, Office of Biological and Environmental Research (award number DE-SC0018409) to T.J.L. and H.K.G.; the Natural Resources Conservation Service of the U.S. Department of Agriculture (award number 96-3A75-16-032) via Ducks Unlimited to T.J.L.; and by a National Science Foundation fellowship (award number DGE-1747503) to S.A.S. Any opinions, findings, and recommendations expressed in this material are those of the authors and do not necessarily reflect the views of the funding organizations.

## Author contributions

T.J.L. designed the study. S.A.S. developed the yield model. M.B., S.A.S., and T.J.L. conducted analyses and visualization. T.J.L. wrote the first draft, and all authors contributed to the interpretation of the results and draft revision.

## Competing interests

The authors declare no competing interests.
