## [Peer Review File · Nature Communications]

Peer Review File - Reviewers' comments first round:

Reviewer #1 (Remarks to the Author):

General comments:

In their submitted paper 'Cropland expansion in the United States produces marginal yields at high costs to wildlife' to Nature Communications, Lark et al. calculate the area of cropland expansion for the years 2008-2016 in the US, based on a LUC approach from previously published papers. The identified expansion areas are further used to quantify the loss of former 'intact prairie'. In a first step, statistical yields on county basis are used to calculate spatial yield distributions by using a random forest approach. Accordingly, yields of expansion areas are calculated and compared to national averages as well as neighboring pixels. Hence, yield differences are found to be lower on large and local scale. It was concluded that expansion areas are less favorable for agricultural production than existing crop areas. Also, it was found that high productive regions have a smaller difference in yields for expansion areas compared to yields on cultivated land compared to regions with a lower production. In a second step, the impact of the identified expansion on Monarch butterfly population and waterfowl breeding habitats, both used for representing wildlife, are investigated for two expansion hot spots: the Prairie Pothole Region, an important wetland area in the USA used by many waterfowl species for breeding, and the Midwest for the Monarch butterflies. Thereby, not the decrease in Monarch butterfly population itself was investigated, but the loss of milkweed stems that are the only food source for the Monarch butterfly larvae. The paper is well written and it provides strong evidence for its conclusions. It is of high interest and importance for a broad range of potential readers from different disciplines and the results are relevant and novel. The conclusions drawn are well argued. The discussion needs to address further points and a more critical reflection about uncertainties and limitations of the approach. A major concern is the accuracy of the LUC assessment, the suitability assessment and the yield calculation. For all three cases, more information should be provided and the approaches should be better and more clearly described, not necessarily longer.

According to Dunn et al. (2015) the CDL data used in this study for LUC detection is not intended to measure LUC. However, I recognized that limitations and recommendations in the use of CDL data are already discussed in Lark et al. (2017). Given also the comparison of different datasets in Supplementary Table 6, there seems to be large differences between different products. These large divergences make it difficult to say anything about the quality of this approach. Additionally, the trends of cropland expansion by FAOSTAT contradicts the data of this study, mainly between 2008 and 2012, when FAOSTAT estimates a net reduction of cropland in the USA, while an expansion peak is described in 2011 according to this study. How does this large divergence come from?

Another major question that is not addressed in the paper is why Monarch butterflies and waterfowls are chosen as the only representatives for wildlife habitats? What about other datasets of biodiversity or endemism richness - such as PREDICT database (<https://www.predicts.org.uk/>), the Global Biodiversity Information Facility (GBIF: <https://www.gbif.org/>), IUCN or BirdLife data? It would be very interesting to instead or in addition consider other indicators of biodiversity declines, such as vertebrates, invertebrates and other plants, that may respond differently to anthropogenic pressure. Thus, it is important to include a wide a range of species (not necessarily in this study, but maybe for further studies). I suggest adding a statement on this issue to the discussion - maybe as an outlook.

For further studies, it would also be interesting e.g. to identify potential land for cropland expansion with low impact on biodiversity and higher production than average. Is this possible? Generally, there is some confusion about time periods, how data was applied or compiled. In order to trust in the robustness of the approach and the results, methods and data must be described more clearly. The applied random forest approach is not explained and it cannot completely be reproduced how yields are calculated. Is there evidence that a random forest approach is a proper

tool for distribution of statistical yield data – also to expanded areas? According to (Jeong et al., 2016) the approach may result in a loss of accuracy when predicting the extreme ends or responses beyond the boundaries of the training data – which would be the case for expansion areas. Also, the agricultural suitability approach (assumptions, what crops are considered, etc.) is not described properly. This leaves some space for open questions. Also, it is not described how was dealt with different spatial resolution of applied datasets?
The author instructions of Nature urge to use SI units. Accordingly, please use km², m² or ha instead of acres.

Overall, I suggest major revisions with the feeling that most of the points raised can be easily addressed by the authors.

Line-specific comments:

Abstract: Please use relative numbers for milkweed stems or even better describe the meaning of 223 mio. milkweed stems, since the meaning of these absolute numbers in the abstract are difficult to understand for at least most of the readers.

Line 44: I suggest to improve the sentence: ... a significant global share of carbon emissions from cropland expansion in the US.

line 74: I think it should be 'time series' instead of 'times series'.

Line 74-75: CDL data is not satellite crop data. It also includes various other input data, such as statistical data and ground truth data from surveys and other ancillary data, such as the National Land Cover Data set. Maybe also important to mention in this context is that the mapping accuracy ranges from 85 - 95% for 2009 (Boryan et al., 2011). Please better describe the data.

Ln 88: It is not wrong, but somehow confusing to read 'in the 8 years following 2008', since the abstract and also Supplementary Table 1 says from 2008-2016, which is a 9-year period including 2008.

Ln 144,145 and Figure 9: A bit confused about wording. In line 148, it is called 'crop suitability', which in my understanding is different to an agricultural suitability.

Ln 153: Was this data applied for 2008-2016?

Ln 165: From 2008 to 2016 is 9 years study period.

Ln 188: I disagree that the findings of this study are 'similar in magnitude' compared to other data. I also cannot see that in Supplementary Table 6. Additionally, as already said in the major comments, other data even show the opposite trends. To me, the large disagreement and possible reasons must be added to the discussion and possible implications on the results must be reflected critically.

Ln 304-306: Citation #81 seems not to be a peer-reviewed paper, which I find critical since it is about the core methodology. Also, this working paper can't be accessed or I can't find it anywhere for download.

Ln 309: What means 'general treatments'. Please describe.

Ln 314,315: Why are the year 2001 and 2006 included, since they are outside of the 2008-2016 period? The next sentence says that the data was combined over the study period, which would exclude all NLCD data, except 2011. It is not clearly described how the NLCD data was further used or combined with the CDL data. It is getting clear after reading line 331, but I suggest mention it before in order to not get confused about that.

Line 330-332: Is this a must or are there exceptions possible - e.g. intermittent cropland within NLCD data or fallow land due to crop rotations between the years 2001, 2006 and 2011?

Line 334: CDL data goes from 2008-2017 (see line 313), which would be 10 years. If it goes from 2008-2016, it would be 9 years. To me, it would make sense to include 2017 in order to be able to guarantee a 2 year remaining of cropland for the last year of the analysis, 2016. Please describe this also more clearly.

Line 340-342: Is there any evidence, that replacing patches smaller than 5 acres with the class of the nearest remaining patch does not lead to classification errors and thus overestimations of expansion areas? Often, smaller patterns in agriculture are used for e.g. grass strips. Wouldn't it be more rational to exclude these areas from the analysis? Also following your explanation until line 351, this would make sense due to higher uncertainties. What is the assumption for applying a nearest neighbor interpolation approach? Please explain. Or can you at least quantify how much an exclusion would change the results? Maybe the impact is not so large.

Line 368: Again, I think that Nature usually does not accept citations of non peer-reviewed material for methods.

Line 372: Citation missing for Brieman 2001. Also not listed in the References!

Line 377: USDA NASS citation missing here, refer to #86.

Line 371ff: Which crops did you consider? Did you distinguish between corn yield and silage yield? If yes, on what basis? Do you distinguish between irrigated and rainfed yields?

Line 375: Is there evidence, that annual climate variables are suitable to use for such an approach? I assume that it would be much better to apply climate data at least only over the growing period, because annual data are not relevant for crop growth expect for perennial crops.

Line 375 and Supplementary Table 9: Please provide spatial resolution for each of the gridded different data listed in the table or add it to the sentence.

Line 377f: This is done for each county individually always taking the extend for each crop of the corresponding county from CDL?

Line 384: I think the applied approach should be explained briefly without reading the (not correctly) cited papers. Briefly describe the approach, not all readers might know it, and do not just say that 'each model was applied'.

Line 389: I am not sure if this simple compilation explained from line 386-388 requires a formula? To save space this could be deleted.

Line 406: According to Supplementary Table 9, slope comes from USGS NED and not from gSSURGO. Are different slope data used for different analysis?

Line 414,415: Spelling error: Penman-Monteith. Also in Supplementary Table 9. What do you mean with reference evapotranspiration? What kind of reference (e.g. grassland reference)? Maybe you mean potential evapotranspiration instead of reference evapotranspiration, because it doesn't make sense to me? Penman-Monteith calculates actual evapotranspiration.

Line 419: 'consider' instead of 'estimate' would make it better to understand in the context that it was additionally included in comparison to Pleasants (2016)?

Line 420: What does CRP stand for? Write out abbreviation (CRP) in full at first use (not in figure captions).

Line 430: not sure, but maybe better: had not been ...

Line 430: Is it legal to assume that land is 'intact' if it has not been used for crop production or pasture/hay production before? I suggest to change wording discuss it.

Line 451: Add reference to Supplementary Table 6.

Line 455: What is the exact definition of cropland used in this study? I can't find it! It would be good to have that in the beginning of the methods.

Literature #17: Journal information is missing, or if this is a monography, City and Publisher information is missing.

Literature # 84: Year or access date missing.

Figure 1: Add abbreviations for States in Supplementary Table 1 and refer to it here. Not all readers are from the US.

Figure 7: Map does not show a), b), c) and d)

Supplement:

Most of the Supplementary Figures contain a short description on the main result and its interpretation of the figure. However, for better understanding the Figures, I suggest to also include data sources and description on how it was compiled. E.g. Supplementary Figure 6: It is not described how crop-specific changes were obtained. Although I assume to understand how this was produced, it should be added to the figure caption (not only results). Also used datasets in figure caption should be cited (compare Supplementary Figure 7).

Supplementary Figure 7: Add abbreviations for States in Supplementary Table 1 and refer to it here. Not all readers are from the US.

Supplementary Figure 8: a) b) c) d) e) f) are not displayed.

Supplementary Table 6: Why are CDL based (this study) numbers not the same than in Supplementary Table 1?

Line 257: Add also personal communications to References.

Line 249-262: Add one sentence of the uncertainty statement to the discussion of the main paper and refer to the supplement for more details.

Line 277: Johnson (2013) is not referred to the References.

Line 331f: Add one sentence on the uncertainty of CDL data also to the discussion of the main paper and refer to the supplement for more details.

Line 482f: What spatial resolution has e.g. the climate data used for the random forest approach

and does it fit to the MMU used in the LUCC analysis?

References:

Boryan C, Yang Z, Mueller R and Craig M 2011 Monitoring US agriculture: the US Department of Agriculture, National Agricultural Statistics Service, Cropland Data Layer Program Geocarto International 26 341-58

Jeong J H, Resop J P, Mueller N D, Fleisher D H, Yun K, Butler E E, Timlin D J, Shim K-M, Gerber J S, Reddy V R and Kim S-H 2016 Random Forests for Global and Regional Crop Yield Predictions PLOS ONE 11 e0156571

Dunn et al. (2015): Comment on 'Cropland expansion outpaces agricultural and biofuel policies in the United States'. Accessible online:
https://www.google.de/url?sa=t&rct=j&q=&esrc=s&source=web&cd=2&cad=rja&uact=8&ved=2ahUKewi_8PLvx7flAhUQKVAKHeNXDdkQFjABegQIBhAC&url=https%3A%2F%2Fgreet.es.anl.gov%2Ffiles%2Fcomments-cropland-expansion&usg=AOvVaw14qVoEUc8YyEXpy8hKnz9P.

Reviewer #2 (Remarks to the Author):

This manuscript presents an ambitious study that uses a comprehensive geospatial dataset characterizing ten years of agricultural land use change to characterize patterns of change and their environmental implications. There is quite a lot of information presented here, which is a strength as well as a limitation. Overall the results are very interesting, important, and policy relevant. However, the paper is not framed well in terms of specific objectives or questions. As a result, it comes across as more of a reference document than an incisive scientific article.

The methods for the land cover change analysis are sound. This component of the study is carefully done and meticulously documented, which is particularly impressive given its national scope. However, some of the secondary analyses of ecological impacts are more speculative and less supported by data. The discussion addresses several interesting points but overall is somewhat lacking in direction and focus, which is likely a reflection of the broad scope of the assessment combined with limited framing in the introduction.

Line 22: Not clear what is mean by "potentially intact" prairie. Is there a more precise term that could be used in the abstract?

Lines 32-34: Can you specify a specific range of years for this "initial timing"?

Line 69: I would like to see a stronger rationale for addressing this time period besides data availability. Can you provide some additional justification for why this is a relevant and important period over which to study agricultural expansion?

Line 80: Here again we see the term "intact grasslands", which will mean different things to different people. The criteria for defining an intact grassland should be provided here where it the term is first used in the paper.

Lines 74-84: The final paragraph of the introduction makes this study seem rather descriptive. This is not necessarily a bad thing, but the paper would be strengthened if the authors could frame some more specific objectives, questions, or hypotheses that drive the analysis.

Figure 1: There is a lot of information in this state-by-state breakdown, but the manuscript doesn't really address the state-by-state differences, just the overall national trends. I think it would be more effective to present some type of regional breakdown with fewer subfigures, and then include the state-level breakdown in the supplements as a reference.

More generally, I found that the presentation of results referenced the supplemental material quite heavily, to the point where I really needed to be looking at the supplement at the same time that I

was reading the paper. I would encourage the authors to rethink and reconfigure the figures and tables in the main article so that they more directly support the narrative of the results. Eliminate extraneous details (e.g., see the previous comments on Figure 1), but try to include figure that directly support all of the most important results.

Line 160: *from* 2008-2016?

Lines 177-178: As noted in the previous comments, this definition of "potentially intact" land should be provided earlier in the paper. Also, I'm somewhat troubled by this definition, as a substantial portion of these lands could have been cultivated and abandoned or converted to hay or pasture more than 25 years ago. On the other hand, it does seem likely that these older grasslands are storing more carbon and may have greater levels of biodiversity than younger grasslands even if they are not truly "intact". Clearly, the 25-year cutoff is imposed by data constraints rather than having an explicit ecological significance. I think it may be more effective to frame this comparison differently as "older" versus "younger" grassland as opposed to the idea of "potentially intact", particularly since we do not really know the proportion of these older grasslands that have never been cultivated, and because the disturbance history of these grasslands likely varies among regions.

Line 183: Following on the preceding comment, terms such as "likely intact" really don't have a clear scientific meaning.

Lines 207-238: I think I agree with most of the points made here in the discussion of yields, and I like the idea of exploring yield differentials at different scales (local and national). But I find myself questioning what the new insights are here. For me, it is already well understood that most of the prime agricultural land is already used for crops, and therefore agricultural expansion almost always has to occur in locations with lower yields. Perhaps including some more direct comparisons with results of previous agricultural land change studies would help to clarify how these results are extending our knowledge of agricultural land systems.

Lines 314-316: Some additional information should be provided on the NLCD change estimates. I assumed that these estimates were derived from the newest (2016) version of the NLCD, which includes a 2008 as well as a 2016 epoch. But here the text implies that NLCD results are based on the 2001, 2006, and 2011 products. Please clarify to avoid confusion.

Lines 418-422: I think that the estimation of conversion effects on milkweed stems was a weak link in the methodology. It is difficult to figure out how these estimates were generated without burrowing deeply into the supplemental materials. After reading the supplement, it is clear that these estimates are based on extrapolations from a relatively small dataset, with considerable (and unmeasurable) uncertainty as a result. I did think it was interesting to contrast the effects of grassland loss with those of GMOs and pesticides in croplands, and I don't disagree with the idea that conversion is likely having a large impact on milkweeds and monarch habitat. But I believe that this results of this assessment need to be presented much more cautiously.

Supplementary Figure 13: The performance of the yield models is impressive. I would be interested in seeing the relative importance of the predictor variables in the yield models for each crop. This information could be added in an extra figure or table in the supplementary material.

Reviewer #3 (Remarks to the Author):

This manuscript tracked cropland expansion during 2008 to 2016 cross CONUS using USDA NASS Cropland Data Layer. The authors then estimate the yield difference between expanded cropland and the benchmark yield, which would be either the national average or local average yield within a 10km window. The authors found that yield 69.5% of new cropland area produced yields lower than the national average, with a mean yield deficit of 6.5% (compared with national average). From those numbers, the authors argued that those new cropland produced "marginal" yields. On the other hand, the authors translated the cropland expansion area into loss of milkweed stems

and nesting opportunities for waterfowl using existing methods. They found a loss of over 223 million milkweed stems in the Monarch butterfly's Midwest summer breeding range, and reduced waterfowl breeding habitat by over 138,000 nesting opportunities in the Prairie Pothole Region of the northern plains. They then argued that those are "high costs" to wildlife.

The manuscript is well-organized and well-written. However, I have the following concerns for the author to consider.

Firstly, I have a feeling that the definition of "marginal yields" and "high costs" are kind of arbitrary and subjective, especially when we considering that the authors were comparing (or hinting a comparison between) some numbers of yield difference (which represent the crop productivity per unit area) with some other numbers of ecological cost (those are calculated for all the converted area). It's just like you are comparing yield with total production. I don't think any potential comparison like this is fair as they lack of similar benchmark. Also, we don't know the exact definition of "marginal yields" and "high costs". I checked the numbers for yield difference, and found the numbers actually differed a lot between using national average yield and using local average yield as benchmark. The authors reported numbers in the abstract using national average yield as benchmark. However, we actually see much smaller number when using local average yield as benchmark. For this yield difference calculation, I think using local average yield is much more reasonable and would suggest the authors use the local average yield only. For the "high costs", it seems there was not any benchmark. We actually don't know how high is really "high costs".

Secondly, the authors used the method described in Pleasants (2016) to calculate the milkweeds lost, while Pleasants used the author's earlier land cover change detection work (Lark et al., 2015). Compared these two earlier work which used data from 2008 to 2012, this work used 4-year more data ranging from 2008 to 2016 with some technical revisions in the data processing workflow. This makes me feel that the innovation of this manuscript is reduced, though I admit that adding the waterfowl breeding habitat assessment component do increase the scope and novelty of this manuscript. Considering that CDL data is also available for 2017 and 2018, I am not sure why the authors excluded those two years.

Thirdly, the ecological assessment lack of uncertainty quantification. I see there is some information uncertainty quantification in Pleasants's work. Would it be possible that the authors conduct similar work for uncertainty? This is important especially considering that those numbers are based on ecological survey and are prone to uncertainties. The authors should also add some discussions on the uncertainty in their estimations.

Other comments:

Yield modeling part: Could you please be more specific on the temporal and spatial resolutions of those predictors shown in supplementary Table 9? Did you use monthly value, yearly mean, or growing season mean values? And how did you aggregate those data into county level? All these details should be clarified.

Response to Reviewers' comments

Dear Reviewers,

Thank you very much for the thoughtful reviews of our manuscript and your helpful suggestions for revision. We greatly appreciate the recommendations and feedback provided throughout and have updated our paper and associated analyses accordingly. Some of the larger changes we made were (1) improving the characterization of our yield model, including expanded description of the methods, explanation and justification of the random forest approach, and addition of variable importance plots, (2) strengthening of the ecological analyses, especially of lost milkweed, by including estimates of uncertainty, assessing the loss in relative (rather than only absolute) measures, and discussing the uncertainty and future research directions, and (3) clarifying the apparent discrepancies with other data such as FAOSTAT estimates, which stem from changes within the FAOSTAT's source data and methodology.

Smaller, across-the-board revisions included the renaming our description of "intact prairie" to "long-term grasslands", improving the consistency in how we reference years of analyses, and adding discussion of uncertainties surrounding the land change and ecological analyses. We also restructured Figure 1 (now Fig. 2) and Supplementary Figure 7 to better capture the regional trends in spatial and temporal conversion patterns, as well as incorporated other minor suggestions throughout. Each of the specific edits has been tracked in the manuscript and is also described in detail in our responses to comments below, which are demarcated by ">>>" and colored blue for consistency. All authors have reviewed and approved the submitted revisions and responses.

Thank you again for your time and effort in reviewing our paper and providing feedback, and we look forward to your additional thoughts and feedback.

Reviewer #1 (Remarks to the Author):

General comments:

In their submitted paper 'Cropland expansion in the United States produces marginal yields at high costs to wildlife' to Nature Communications, Lark et al. calculate the area of cropland expansion for the years 2008-2016 in the US, based on a LUC approach from previously published papers. The identified expansion areas are further used to quantify the loss of former 'intact prairie'. In a first step, statistical yields on county basis are used to calculate spatial yield distributions by using a random forest approach. Accordingly, yields of expansion areas are calculated and compared to national averages as well as neighboring pixels. Hence, yield differences are found to be lower on large and local scale. It was concluded that expansion areas are less favorable for agricultural production than existing crop areas. Also, it was found that high productive regions have a smaller difference in yields for expansion areas compared to yields on cultivated land compared to regions with a lower production. In a second step, the impact of the identified expansion on Monarch butterfly population and waterfowl breeding habitats, both used for representing wildlife, are investigated for two expansion hot spots: the Prairie

Pothole Region, an important wetland area in the USA used by many waterfowl species for breeding, and the Midwest for the Monarch butterflies. Thereby, not the decrease in Monarch butterfly population itself was investigated, but the loss of milkweed stems that are the only food source for the Monarch butterfly larvae.

The paper is well written and it provides strong evidence for its conclusions. It is of high interest and importance for a broad range of potential readers from different disciplines and the results are relevant and novel. The conclusions drawn are well argued. The discussion needs to address further points and a more critical reflection about uncertainties and limitations of the approach.

A major concern is the accuracy of the LUC assessment, the suitability assessment and the yield calculation. For all three cases, more information should be provided and the approaches should be better and more clearly described, not necessarily longer.

>>>Thank you for the very thoughtful and complete review. We have carefully considered each of the comments and made revisions to the manuscript, accordingly. These edits are detailed below, with particular attention to the LUC assessment, suitability assessment, and yield calculations.

According to Dunn et al. (2015) the CDL data used in this study for LUC detection is not intended to measure LUC. However, I recognized that limitations and recommendations in the use of CDL data are already discussed in Lark et al. (2017). Given also the comparison of different datasets in Supplementary Table 6, there seems to be large differences between different products. These large divergences make it difficult to say anything about the quality of this approach. Additionally, the trends of cropland expansion by FAOSTAT contradicts the data of this study, mainly between 2008 and 2012, when FAOSTAT estimates a net reduction of cropland in the USA, while an expansion peak is described in 2011 according to this study. How does this large divergence come from?

>>>Thanks for these comments and question. Indeed, the concerns of Dunn et al. (2015) were largely discussed in Lark et al. (2017), and although those recommendations are not covered in full here, they are embodied within our new dataset and analyses.

>>>We have made several adjustments to our manuscript to address the differences in datasets in Supplementary Table 6, and these are further described in our response to the specific comment about that data in the minor comments section below.

>>>The FAOSTAT data relies upon an aggregated definition of cropland from the USDA Census of Agriculture in such a way that unfortunately does not provide a reliable estimate of active cropland area for comparison with our study. In particular, the FAOSTAT estimate is based on the measure of “total cropland” area from the USDA Census of Agriculture. This metric of total cropland includes the sub-categories of idle cropland as well as cropland-pasture. Inclusion of the cropland-pasture category is particularly problematic, both because it represents land actively used as pasture (rather than cropland) and because the category has undergone changes in definition and administration within the Census’s survey instrument over time (For a good description, see pg 15 of this USDA ERS publication). As such, the decline in “total cropland” as measured by the USDA Census of Agriculture (and reflected by

FAOSTAT) largely represents the decline in the cropland-pasture category, which is largely attributable to the survey questionnaire changes. Given the prevalence and familiarity of many readers with FAOSTAT, we have now added a full paragraph describing this discrepancy to the supplementary text (lines ~291 – 301). We also point to this in a new penultimate sentence of the main text: “Note that we did not compare to data reported by the United Nation’s FAOSTAT database as those metrics are based upon USDA data including the Census of Agriculture’s estimates of total cropland (see **Supplementary note on comparison to other data**).”

Another major question that is not addressed in the paper is why Monarch butterflies and waterfowls are chosen as the only representatives for wildlife habitats? What about other datasets of biodiversity or endemism richness - such as PREDICT database (<https://www.predicts.org.uk/>), the Global Biodiversity Information Facility (GBIF: <https://www.gbif.org/>), IUCN or BirdLife data? It would be very interesting to instead or in addition consider other indicators of biodiversity declines, such as vertebrates, invertebrates and other plants, that may respond differently to anthropogenic pressure. Thus, it is important to include a wide a range of species (not necessarily in this study, but maybe for further studies). I suggest adding a statement on this issue to the discussion - maybe as an outlook.

>>>Thank you for these great suggestions! We initially selected Monarchs and waterfowl for their high level of interest to the public, conservation organizations, and policymakers—three of our targeted audiences—and have added text explaining these choices to the introduction: “These specific taxa were selected for their public familiarity, representation of broad wildlife types, and recognition as broader indicator species²³.” We also agree that further studies investigating the wildlife impacts would be insightful and important, and have added this as an outlook in the discussion with reference to some of the noted helpful data sources: “Collectively, our analyses of representative pollinator, waterfowl, and native plant habitats show that cropland expansion is infringing upon high quality natural land with the potential to disproportionately affect the wildlife that depend upon it. Given these indications, future analyses might aim to further examine the impacts of expansion on a wider range of taxa^{59,60} – especially endemic species – and how they manifest throughout the population, community, and ecosystem levels. Such biodiversity impacts could also be compared directly with other tradeoffs, such as effects on climate, water use, and water quality.”

For further studies, it would also be interesting e.g. to identify potential land for cropland expansion with low impact on biodiversity and higher production than average. Is this possible?

>>>Agreed. We have performed some preliminary explorations of the tradeoffs of expanding into remaining uncultivated land and hope to further pursue this in future work, perhaps in comparison of yields, carbon, and biodiversity.

Generally, there is some confusion about time periods, how data was applied or compiled. In order to trust in the robustness of the approach and the results, methods and data must be described more clearly. The applied random forest approach is not explained and it cannot completely be reproduced how yields are calculated. Is there evidence that a random forest approach is a proper tool for

distribution of statistical yield data – also to expanded areas? According to (Jeong et al., 2016) the approach may result in a loss of accuracy when predicting the extreme ends or responses beyond the boundaries of the training data – which would be the case for expansion areas. Also, the agricultural suitability approach (assumptions, what crops are considered, etc.) is not described properly. This leaves some space for open questions. Also, it is not described how was dealt with different spatial resolution of applied datasets?

>>>Thanks for these comments. We have addressed and describe our changes to each of these in response to the associated specific comments below.

The author instructions of Nature urge to use SI units. Accordingly, please use km², m² or ha instead of acres.

>>>Thanks for flagging this. We made a special request to use the unit of acres in our submission letter to the editor, but we realize that this was not available to reviewers. Because of the specific intended policy applications of this research (e.g. U.S. Farm Bill, USDA policies, and Renewable Fuel Standard), we would like to use a unit of “acres” to report area throughout the manuscript in order to maintain consistency with the relevant federal agencies’ documentation and the units referenced in other US data and policymaking processes. However, we have added a statement of unit equivalence – “(1 acre = 0.40 hectares)” after the first incidence of the unit ‘acres’ in the opening sentence of the results section.

Overall, I suggest major revisions with the feeling that most of the points raised can be easily addressed by the authors.

Line-specific comments:

Abstract: Please use relative numbers for milkweed stems or even better describe the meaning of 223 mio. milkweed stems, since the meaning of these absolute numbers in the abstract are difficult to understand for at least most of the readers.

>>>Great suggestion. We have updated the abstract (and the associated main text and figure 4) to reference relative numbers for milkweed stems. The new abstract sentence now states “Observed conversion infringed upon high-quality habitat that, relative to unconverted land, had provided over three times higher milkweed stem densities in the Monarch butterfly’s Midwest summer breeding range...”

Line 44: I suggest to improve the sentence: ... a significant global share of carbon emissions from cropland expansion in the US.

>>>We’ve maintained the sentence as is, out of concern that readers could misinterpret a “significant global share” as a major/majority proportion of global emissions.

line 74: I think it should be 'time series' instead of 'times series'.

>>>Good catch. Updated accordingly.

Line 74-75: CDL data is not satellite crop data. It also includes various other input data, such as statistical data and ground truth data from surveys and other ancillary data, such as the National Land Cover Data set. Maybe also important to mention in this context is that the mapping accuracy ranges from 85 - 95% for 2009 (Boryan et al., 2011). Please better describe the data.

>>>We have updated our description of the CDL to correct the satellite misnomer, such that it now states “We began by tracking field-level changes throughout the full time series of nationwide USDA cropland maps²³.” We have also added the following sentences at the beginning of the supplementary methods (lines 381-385) to better describe the data and its accuracy: “We used the USDA Cropland Data Layer (CDL) as the primary input for detecting land conversion. The CDL is an annual 30m resolution, crop-specific land cover map that provides coverage for all states in the conterminous U.S. beginning in 2008, with crop-specific accuracies generally ranging from 85-95%¹⁹.”

Ln 88: It is not wrong, but somehow confusing to read 'in the 8 years following 2008', since the abstract and also Supplementary Table 1 says from 2008-2016, which is a 9-year period including 2008.

>>> In the preceding paragraph, we updated our language from “...an eight year period” to “...we identified changes over eight conversion years (e.g. 2008-09 = one conversion year).” We think that this change should also help set-up and clarify the Ln 88 reference of “in the eight years following 2008”.

Ln 144,145 and Figure 9: A bit confused about wording. In line 148, it is called 'crop suitability', which in my understanding is different to an agricultural suitability.

>>>We have updated both the main text and supp. fig. 9 to now state “cultivation suitability,” which is the specific term used by the USDA in their documentation of the land capability classification.

Ln 153: Was this data applied for 2008-2016?

>>> Yes, it represents conditions during our study period, and we have updated the text to specifically reference “during our study period” to help make this clear.

Ln 165: From 2008 to 2016 is 9 years study period.

>>>We changed “From 2008 to 2016” to “during our study period”, such that it references the previously defined dates and duration to remove any discrepancy and improve consistency throughout.

Ln 188: I disagree that the findings of this study are 'similar in magnitude' compared to other data. I also cannot see that in Supplementary Table 6. Additionally, as already said in the major comments, other data even show the opposite trends. To me, the large disagreement and possible reasons must be added to the discussion and possible implications on the results must be reflected critically.

>>>Thanks for this fair assessment. We agree with your perspective and have updated our interpretation of the data accordingly. We have changed the main text from “similar in magnitude” to “similar in direction” to better reflect this. We have also updated the text in the caption for Supplementary Table 6 to mention agreement only in *net* cropland expansion, and to remove reference to agreement regarding the *extent or magnitude* of such expansion. The new description states “The four national products compared here—the NLCD, the Census of Agriculture, the NRI, and our study—all report annual average rates of net cropland expansion of between 822,000 and 1.39 million acres, representing ~~relative congruence on the extent~~ a consensus of net cropland expansion during the last decade in the U.S.”

>>>As noted regarding the major comment above, we also added the following paragraph to the supplemental material and a references sentence to the main text to discuss and clarify the apparent large disagreement with other data (FAOSTAT): “Other potential comparison datasets include national estimates from the United Nations’ FAOSTAT database¹² and the USDA NASS annual surveys¹³. Data from the FAO regarding arable land and cropland extent for the US is based upon USDA Census of Agriculture estimates for “total cropland” in the US. However, this broad USDA classification also includes subcategories of idle cropland and cropland-pasture, thereby cushioning its estimate of active cropland extent. Furthermore, there have been shifts to the definition and presentation of the cropland-pasture category within the Census of Agriculture survey instrument over time¹⁴. These changes have led to discontinuity in land’s classification as either cropland or pasture across time, thereby further muddling the use of USDA Census of Agriculture estimates of total cropland and the associated FAOSTAT data points as indicators for active cropland extent. As such, we compared to only the specific categories in the USDA Census of Agriculture that best reflect active cropland extent—the sum of planted, failed, and fallow cropland—rather than the aggregated metric of total cropland reported in the Census and reflected by the FAO data.”

Ln 304-306: Citation #81 seems not to be a peer-reviewed paper, which I find critical since it is about the core methodology. Also, this working paper can't be accessed or I can't find it anywhere for download.

>>>We have removed this redundant citation (to an unpublished dissertation chapter), as each of the referenced methods are included in the other, peer-reviewed work cited.

Ln 309: What means 'general treatments'. Please describe.

>>>We intended to reference treatments that were not class-specific, such as the spatial filter and minimum mapping unit. We have updated the text from “general” to “non-specific” to clarify, such that this overview sentence now states “...refining data via class-specific and non-specific treatments...”. The specific and non-specific treatments are further described in the subsequent paragraphs.

Ln 314,315: Why are the year 2001 and 2006 included, since they are outside of the 2008-2016 period? The next sentence says that the data was combined over the study period, which would exclude all NLCD data, except 2011. It is not clearly described how the NLCD data was further used or combined

with the CDL data. It is getting clear after reading line 331, but I suggest mention it before in order to not get confused about that.

>>>Good catch. We removed the phrase “over the study period” from the subsequent sentence in order to correct this misdescription. We indeed use some data from outside the 2008-16 study period (2001 and 2006 NLDC, and 2017 CDL) for additional context -- e.g. determining whether fields have been cropped in the recent past -- to improve classification within our study period.

Line 330-332: Is this a must or are there exceptions possible - e.g. intermittent cropland within NLCD data or fallow land due to crop rotations between the years 2001, 2006 and 2011?

>>>As worded here, it is describing only areas “converted to cropland” and it is a true “must”. Areas with intermittent cropland within the NLCD would instead be classified within our intermittent cropland category and are explicitly excluded from our estimates of conversion area. Our classes of intermittent cropland and stable cropland each have different rules (with more exceptions) and these are described in detail in the supplementary methods.

Line 334: CDL data goes from 2008-2017 (see line 313), which would be 10 years. If it goes from 2008-2016, it would be 9 years. To me, it would make sense to include 2017 in order to be able to guarantee a 2 year remaining of cropland for the last year of the analysis, 2016. Please describe this also more clearly.

>>>Good point. We indeed included 2017 in our analysis in order to ensure 2 years of cropland after a conversion. We have updated the text and changed “eight years” to “ten years” to correctly reflect the total number of CDL years considered in this determination.

Line 340-342: Is there any evidence, that replacing patches smaller than 5 acres with the class of the nearest remaining patch does not lead to classification errors and thus overestimations of expansion areas? Often, smaller patterns in agriculture are used for e.g. grass strips. Wouldn't it be more rational to exclude these areas from the analysis? Also following your explanation until line 351, this would make sense due to higher uncertainties. What is the assumption for applying a nearest neighbor interpolation approach? Please explain. Or can you at least quantify how much an exclusion would change the results? Maybe the impact is not so large.

>>> Great question. While we did not perform a quantitative sensitivity analysis on the effects of different patch sizes and replacement methods, we did qualitatively investigate the effects of different approaches by visually comparing results to very high resolution aerial imagery to guide our selection.

Removing and not replacing patches smaller than 5 acres in size would have resulted in their overall exclusion from the analyses. This was not a desirable approach because estimates of the total area of each land transition type (stable cropland, conversion, intermittent, etc.) would be underestimated, and certain land transition types (e.g. small patches of conversion) could be systematically underrepresented. We thereby choose to replace small, uncertain patches rather than exclude them.

To select a method for replacement, we decided to use a nearest neighbor approach based on Tobler's first law of geography (everything is related to everything else, but near things are more related than distant things). The nearest neighbor approach is also a relatively common approach in land use modeling.

We have added this justification and associated reference to the main text methods, such that it now states "Contiguous patches less than five acres in size were removed, and the resulting void pixels replaced with the transition class of the nearest remaining patch based on Tobler's first law of geography – that near things are more similar than distant ones.^{80,81}"

We have also added the following full explanation of this step to the supplementary methods to provide further information: "This MMU involved removing patches of broad LUC smaller than five acres and replacing them with the trajectories (and associated LUC classes) of the nearest pixel neighbors. Without replacement, the total area of land and each broad LUC class would be underestimated, and certain LUC types (e.g. small patches of conversion) could be systematically underrepresented. To perform the replacement, we filled the voided pixels using a nearest neighbor approach based on the Tobler's first law of geography¹⁵."

Line 368: Again, I think that Nature usually does not accept citations of non peer-reviewed material for methods.

>>>We have retained this citation for the time being and will look to the editorial team for guidance on its acceptability. To help with evaluation, we have added the ISBN number and permalink to the publication. Should the citation need to be removed, we have included an alternative citation that supports the broader general sentence and would remove the specific finding referenced from this material (">98% for all years").

Line 372: Citation missing for Brieman 2001. Also not listed in the References!

>>>Thanks for catching this. We've now added it to both the text and the references.

Line 377: USDA NASS citation missing here, refer to #86.

>>>Citation added.

Line 371ff: Which crops did you consider? Did you distinguish between corn yield and silage yield? If yes, on what basis? Do you distinguish between irrigated and rainfed yields?

>>>We have substantially revised the methods section to clarify these questions. We updated the text to say: "to predict the expected average grain yields of corn, soybeans, and wheat". We further clarified that we did not explicitly distinguish between wheat varieties. We also did not distinguish between irrigated and non-irrigated yields. Instead, as we now describe more thoroughly in the methods, this agnosticism towards a field's irrigation status enables our models to predict a representative yield based

on the most common management practices used within a given biophysical setting. To make this clearer we have included a more thorough description of how our yield estimates should be interpreted and contrasted them from other, more traditional yield modeling approaches.

Line 375: Is there evidence, that annual climate variables are suitable to use for such an approach? I assume that it would be much better to apply climate data at least only over the growing period, because annual data are not relevant for crop growth expect for perennial crops.

>>>The approach predicts a “representative yield” based exclusively on the biophysical setting. It is not meant to predict the actual yield from year to year – such predictions would more accurately be made with a remotely sensed approach or process-based simulations. Both of those approaches, though, require a lengthy record of observations, which unfortunately does not exist for newly converted croplands. As such we needed an approach that allows us to make predictions based exclusively on static but fine-scale biophysical covariates. We have updated the text to make this objective clearer, described in the new second paragraph of the yield modeling methods:

“We used crop-specific random forest (RF) models to predict representative grain yields of corn, soybeans, and wheat on newly converted croplands using a suite of biophysical covariate grids. Unlike traditional yield modelling methods that often require a lengthy observational record (e.g. remote sensing or process-based approaches), our method allows for the assessment of likely yields of new fields which may lack such a record. Moreover, our objective was not to estimate yields from year-to-year with unprecedented accuracy or in-season timeliness as is often the goal of traditional approaches but rather to broadly assess the relative production potential of new croplands in relation to those that already existed. Our models therefore do not account for dynamic factors like stochastic variation in weather, anomalous management, or genetic improvements. Their predictions instead intended to represent average expected yields within the period of our training data (2008-2017) and are a function of the local biophysical setting and the management practices implicitly associated with those conditions. For the sake of methodologically consistent comparison, we applied our models to both newly converted and stable pre-existing cropland classes and report only relative (%) differences to ensure proper interpretation of the model’s predictions.”

Line 375 and Supplementary Table 9: Please provide spatial resolution for each of the gridded different data listed in the table or add it to the sentence.

>>>We now include this information in both the revised main text methods and in the caption of the supplementary table. In the main text, it is described in the third paragraph on yield modeling: “TerraClimate grids had a native spatial resolution of 2.5 arc minutes and were resampled to 30m prior to tabulation using the bilinear method in GEE to match the resolution of the CDL and the NCCPI grids. Grids derived from the NED had a 1/3 arc second resolution and were similarly aggregated in GEE to a 30m resolution. Tabulated covariate statistics were then joined to the corresponding yield data to complete the training set.

Line 377f: This is done for each county individually always taking the extend for each crop of the corresponding county from CDL?

>>>Correct, and we have revised the text to make this clearer (see next response).

Line 384: I think the applied approach should be explained briefly without reading the (not correctly) cited papers. Briefly describe the approach, not all readers might know it, and do not just say that 'each model was applied'.

>>>We have substantially revised the text to clarify how the approach was applied. The citation in question was in reference to the computing environment (Google Earth Engine) in which the random forest models were ultimately implemented to map expected yields. The associated text has been updated to clarify this reference and to include additional references where appropriate. The newly added methods descriptions are:

“Random forest is a non-parametric, data-driven method that generates predictions based on an ensemble of bootstrapped classification or regression trees⁹² and has been successfully used by others for yield predictions⁹³ akin to ours. We developed three separate RF models (one each for corn, soybeans and wheat) using training datasets we collated from annual county-level yield averages for each of the three crops and the corresponding means of biophysical covariates within each crop’s planted extent in a given year and county. We used ten years (2008-2017) of county-level crop yield averages from the USDA’s Agricultural Resource Management Surveys⁹⁴ – years in which we could precisely determine the planted location of each crop in each county using the CDL. We retained yield data for all U.S. counties for which it was available and did not differentiate data based on irrigation status nor, in the case of wheat yields, among wheat varieties (e.g. spring, winter, or durum).

Line 389: I am not sure if this simple compilation explained from line 386-388 requires a formula? To save space this could be deleted.

>>>For clarity, we have retained the formula for now, pending further feedback regarding space.

Line 406: According to Supplementary Table 9, slope comes from USGS NED and not from gSSURGO. Are different slope data used for different analysis?

>>>Great point and question. In our first draft we did indeed used two different slope data sets for different analyses (one for the yield model covariate, and one to assess the slope gradient of new croplands). We have now revised our slope gradient analysis such that our paper now only relies on slope as reported by the NED. This did not substantially change the slope gradient results, and it improved consistency across the analyses.

Line 414,415: Spelling error: Penman-Monteith. Also in Supplementary Table 9. What do you mean with reference evapotranspiration? What kind of reference (e.g. grassland reference)? Maybe you mean

potential evapotranspiration instead of reference evapotranspiration, because it doesn't make sense to me? Penman-Monteith calculates actual evapotranspiration.

>>>Thank you for catching this. We have corrected the spelling of Penman-Monteith accordingly. We did not calculate a new index based on reference or potential evapotranspiration, but instead simply use the existing gridded climate water deficit data from TerraClimate and cite this dataset accordingly. To improve clarity of their method, we have updated the text to “Mean annual climate water deficit (2008-2017) was calculated using monthly grids from Terraclimate²⁵, wherein climate water deficit is defined as the difference between a reference evapotranspiration (in this case, potential evapotranspiration) calculated using the Penman-Monteith approach and actual evapotranspiration.”

Line 419: 'consider' instead of 'estimate' would make it better to understand in the context that it was additionally included in comparison to Pleasants (2016)?

>>>Changed accordingly.

Line 420: What does CRP stand for? Write out abbreviation (CRP) in full at first use (not in figure captions).

>>>We now write out Conservation Reserve Program (CRP) in this location.

Line 430: not sure, but maybe better: had not been ...

>>>Changed accordingly.

Line 430: Is it legal to assume that land is 'intact' if it has not been used for crop production or pasture/hay production before? I suggest to change wording discuss it.

>>>Agreed. We've changed the wording of 'intact' to 'long-term' throughout the paper.

Line 451: Add reference to Supplementary Table 6.

>>>Added.

Line 455: What is the exact definition of cropland used in this study? I can't find it! It would be good to have that in the beginning of the methods.

>>>We've now added the following definition to the start of the second paragraph of the methods section, and also point to the table which includes the specific CDL classes that fall within our definition: “Cropland was broadly defined as any area planted to cultivated row, closely grown, or horticultural crops and included cultivated fallow and alfalfa (**Supplementary Table 8**).”

Literature #17: Journal information is missing, or if this is a monography, City and Publisher information is missing.

>>>This citation has been replaced with the appropriate peer-reviewed journal articles for the referenced data.

Literature # 84: Year or access date missing.

>>>We have added the following details: "Version 2.3.2, accessed 11/15/2018"

Figure 1: Add abbreviations for States in Supplementary Table 1 and refer to it here. Not all readers are from the US.

>>>To simplify this for readers, we have replaced the abbreviations from this figure with full state names. To help incorporate a suggestion from reviewer 2, we have also re-arranged the order of each state's graph such that it is generally reflective of the state's location within the U.S. rather than alphabetically sorted, thereby further facilitating state identification and ease of use for all readers.

Figure 7: Map does not show a), b), c) and d)

>>>Good catch. Updated to include subfigure letter labels.

Supplement:

Most of the Supplementary Figures contain a short description on the main result and its interpretation of the figure. However, for better understanding the Figures, I suggest to also include data sources and description on how it was compiled. E.g. Supplementary Figure 6: It is not described how crop-specific changes were obtained. Although I assume to understand how this was produced, it should be added to the figure caption (not only results). Also used datasets in figure caption should be cited (compare Supplementary Figure 7).

>>>We have now included data sources and brief method descriptions where fitting, as follows:

For Supplementary Figure 4, we added "The map displays the percent of the landscape within 3 km x 3 km visualization units that was converted to cropland from grasslands (a), shrublands (b), forests (c), and wetlands (d) between 2008 and 2016. Land cover type derived from the Cropland Data Layer¹ based on the trajectory analysis of conversion and the latest non-crop class prior to a conversion."

For Supplementary Figure 6, we added "For each area of land converted to crop production, the first crop type was extracted from the Cropland Data Layer¹ for the first growing season following a conversion."

For Supplementary Figure 7, we added “For each area of land converted to crop production, the first crop type was extracted from the Cropland Data Layer¹ for the first growing season following a conversion—i.e. land converted between 2008 and 2009 are reported as a 2009 conversion to the crop present in 2009.”

Supplementary Figure 7: Add abbreviations for States in Supplementary Table 1 and refer to it here. Not all readers are from the US.

>>>Like main Figure 1 (now Fig 2), we have replaced the abbreviations in Supplementary Figure 7 with full state names. We have also re-arranged the order of each state’s graph such that it is reflective of the state’s general location within the U.S. rather than alphabetically sorted, thereby further facilitating state identification and interpretation by all readers.

Supplementary Figure 8: a) b) c) d) e) f) are not displayed.

>>>Updated to include subfigure letter labels.

Supplementary Table 6: Why are CDL based (this study) numbers not the same than in Supplementary Table 1?

>>>Thanks for noticing this. We had previously tabulated values in Supplementary Table 1 using a state boundary map from USDA NASS that had simplified boundary geometry. This truncated or skewed some peripheral regions, leading to the small discrepancy in the nationwide values. We have now updated the tabulations using a new state map from the U.S. Census which does not simplify shapes as much, such that the values in Supplementary Tables 6 and 1 now match. We also rounded the Supplementary Table 1 numbers to reflect a more appropriate level of precision.

Line 257: Add also personal communications to References.

>>>The content of one of these communications has since been published, and is now cited as such. The other has been added to the references.

Line 249-262: Add one sentence of the uncertainty statement to the discussion of the main paper and refer to the supplement for more details.

>>>We have added the following sentence from our uncertainty statement to the main text: “Our results embody significant uncertainty, however, and recent field surveys suggest milkweed concentrations of converted grasslands may be even greater than estimated here²⁷ (see **Supplementary note on milkweed conversion**).”

Line 277: Johnson (2013) is not referred to the References.

>>>Added to the references.

Line 331f: Add one sentence on the uncertainty of CDL data also to the discussion of the main paper and refer to the supplement for more details.

>>>We added the following underlined sentence to paragraph 3 of the discussion “The magnitude of our estimates of gross conversion are generally more conservative than others, due in part to our prioritization of improved map confidence in converted areas over complete capture of all conversion (see **Supplementary note on comparison to other results**). The uncertain nature of the underlying CDL and NLCD data may further contribute to observed dissimilarities. For example, it can be challenging to...”

Line 482f: What spatial resolution has e.g. the climate data used for the random forest approach and does it fit to the MMU used in the LUCC analysis?

>>>In our revised methods description of the random forest approach, we’ve now included the spatial resolution of all the covariate layers. Only the climate grids have a spatial resolution greater than the MMU of our LUCC data. However, the climate data was resampled to match the LUCC. Because climate variables generally represent a true gradient, we assume this discrepancy in resolution is minimally important as the climate data ultimately serve to describe broad-scale biophysical gradients. Finer-scale controls on potential yields are captured by the NCCPI layers which is an index that aggregates numerous mapped variables associated with crop performance and has been widely related to map yields (e.g. Egli & Hatfield 2014, Bandaru et al. 2014, and Meehan et al. 2013). The NCCPI has a 30m spatial resolution that matches that of our LUC data and is visually evident in the fine scale spatial variation of our mapped yield predictions. We found that the performance of our model was greatly improved by factoring in climatic gradients, and thus have incorporated them accordingly. The relative importance of including the broad climate gradients is also illustrated in the variable importance plots that we have now added to the supplement (Supplementary Figure 15).

References:

Boryan C, Yang Z, Mueller R and Craig M 2011 Monitoring US agriculture: the US Department of Agriculture, National Agricultural Statistics Service, Cropland Data Layer Program Geocarto International 26 341-58

Jeong J H, Resop J P, Mueller N D, Fleisher D H, Yun K, Butler E E, Timlin D J, Shim K-M, Gerber J S, Reddy V R and Kim S-H 2016 Random Forests for Global and Regional Crop Yield Predictions PLOS ONE 11 e0156571

Dunn et al. (2015): Comment on ‘Cropland expansion outpaces agricultural and biofuel policies in the United States’. Accessible online:

https://www.google.de/url?sa=t&rct=j&q=&esrc=s&source=web&cd=2&cad=rja&uact=8&ved=2ahUKEwi_8PLvx7fIAhUQKVAKHeNXDdkQFjABegQIBhAC&url=https%3A%2F%2Fgreet.es.anl.gov%2Ffiles%2Fcomments-cropland-expansion&usg=AOvVaw14qVoEUc8YyEXpy8hKnz9P.

Reviewer #2 (Remarks to the Author):

This manuscript presents an ambitious study that uses a comprehensive geospatial dataset characterizing ten years of agricultural land use change to characterize patterns of change and their environmental implications. There is quite a lot of information presented here, which is a strength as well as a limitation. Overall the results are very interesting, important, and policy relevant. However, the paper is not framed well in terms of specific objectives or questions. As a result, it comes across as more of a reference document than an incisive scientific article.

The methods for the land cover change analysis are sound. This component of the study is carefully done and meticulously documented, which is particularly impressive given its national scope. However, some of the secondary analyses of ecological impacts are more speculative and less supported by data. The discussion addresses several interesting points but overall is somewhat lacking in direction and focus, which is likely a reflection of the broad scope of the assessment combined with limited framing in the introduction.

>>>Thank you for the helpful big-picture feedback and suggestions on how to improve the paper. The points regarding framing of the study are well-taken, and we have aimed to address these in the revisions. In particular, we added additional context and motivation in the introduction, we listed our specific research questions and focus in the methods, and we added a new topic sentence or revised the current one in most of the discussion paragraphs in order to better connect and organize the section. These changes have been detailed below.

Line 22: Not clear what is mean by “potentially intact” prairie. Is there a more precise term that could be used in the abstract?

>>>We agree that this previous terminology was problematic. We have now replaced the term “potentially intact prairie” with “long-term grasslands” within the abstract and throughout the paper. This new term removes the vagueness surrounding “potentially intact” and is more directly connected to the explicit definition of the assessed lands, which is based on their length of existence.

Lines 32-34: Can you specify a specific range of years for this “initial timing”?

>>>We have now added the text “(~2007 – 2012)” to define this approximate range.

Line 69: I would like to see a stronger rationale for addressing this time period besides data availability. Can you provide some additional justification for why this is a relevant and important period over which to study agricultural expansion?

>>> To better justify the selection and importance of the time period, we have expanded the following sentence and moved it to a new section 4.1 *Study Design*: “Our study time frame of 2008-2016 corresponds with the availability of crop-specific land use data²² and encompasses a variety of market

and environmental conditions—including both high and low crop prices as well as drought, normal, and wet years—thereby providing insights into the more persistent characteristics of cropland expansion across time.”

Line 80: Here again we see the term “intact grasslands”, which will mean different things to different people. The criteria for defining an intact grassland should be provided here where it the term is first used in the paper.

>>>Agreed. We have replaced this use of the problematic term “intact grasslands” with “long-term grasslands”. Given the more literal nature of the new term, we have not defined it immediately here in the study overview, but instead define it at the start of the relevant results section where its application and widespread use first appear.

Lines 74-84: The final paragraph of the introduction makes this study seem rather descriptive. This is not necessarily a bad thing, but the paper would be strengthened if the authors could frame some more specific objectives, questions, or hypotheses that drive the analysis.

>>>Thanks for this helpful suggestion. We have now added the following specific research questions to a new subsection regarding study design, located at the start of the methods section: “Our study was designed and conducted to answer the following three research questions: 1.) What are the annual spatiotemporal patterns of recent cropland expansion, particularly following its resurgence at the end of the 2000s decade? 2.) How do the yields of new croplands compare to those of existing croplands, and how do these differences vary across space and scales? and 3.) What are the absolute and relative impacts of recent cropland expansion on wildlife habitat of public concern? To address these questions, we paired analyses of land use change with those of crop yields and habitat quality to assess the tradeoffs of recent land conversion on both agricultural and natural ecosystems.”

>>>In addition, we have revised the penultimate paragraph of the introduction and added the following text to try to better frame the study: “Collectively, the uncertainty surrounding cropland expansion’s persistence, yields, and implications for wildlife has encumbered meaningful evaluation of its merits and consequences. Given the geographies and characteristics of new croplands and converted habitat, we hypothesize that contemporary US cropland expansion may provide only marginal production gains at significant costs to wildlife. To assess this hypothesis, we mapped cropland expansion...”

Figure 1: There is a lot of information in this state-by-state breakdown, but the manuscript doesn’t really address the state-by-state differences, just the overall national trends. I think it would be more effective to present some type of regional breakdown with fewer subfigures, and then include the state-level breakdown in the supplements as a reference.

>>>We agree that the previous figure provided a lot of information in a format that was not conducive to assimilation or interpretation of broader regional and national trends. To address this, we replaced Figure 1 with a new figure (now Figure 2) in which the state-level results are generally arranged

according to their geographic location and relationships. This helps ease the recognition and observation of broader regional and spatial trends. In addition, we filled in the space below each data series in the line graphs, such that the area below the curves reflects the total area of land conversion. This update further facilitates reading of the figure and its conveyance of spatial-temporal trends, such that regions with large areas under their curves represent areas with large amounts of cropland expansion or abandonment.

More generally, I found that the presentation of results referenced the supplemental material quite heavily, to the point where I really needed to be looking at the supplement at the same time that I was reading the paper. I would encourage the authors to rethink and reconfigure the figures and tables in the main article so that they more directly support the narrative of the results. Eliminate extraneous details (e.g., see the previous comments on Figure 1), but try to include figure that directly support all of the most important results.

>>Thanks for this suggestion. We have revamped the figures in the main text to better support the narrative of the results. We did keep the references to supplemental material to help guide specialized readers to those specific results, though we consolidated the references in certain locations to reduce narrative interruption. Key changes to the main figures include:

-Revision to Figure 1 (now Fig 2), as discussed above, which better communicates the spatial and temporal trends in conversion and thus reduces the need to reference Supplemental Figures 1-3.

-Revision to Figure 4 such that it more directly supports the key results of the paper. Specifically, we now map (in a two-part figure) the relative loss of milkweed stems per county and the relative density of milkweed stems on converted lands compared to unconverted lands. This figure more directly supports the conclusion that higher-quality habitat is being disproportionately converted. It also now does not require readers to reference the associated Supplemental Figures 12 and 13 to understand the spatial factors underlying Figure 4.

-Revision to Figure 7, such that it now normalizes results across all four datasets by comparing the annual (rather than total) cropland expansion within each dataset. This update should reduce the need to reference Supplementary Table 6 in order to get an apples-to-apples comparison.

Line 160: *from* 2008-2016?

>>>To improve the consistency of date references throughout the paper, we have updated this text to now say “during our study period”, which is more fully defined at the start of the paper.

Lines 177-178: As noted in the previous comments, this definition of “potentially intact” land should be provided earlier in the paper. Also, I’m somewhat troubled by this definition, as a substantial portion of these lands could have been cultivated and abandoned or converted to hay or pasture more than 25 years ago. On the other hand, it does seem likely that these older grasslands are storing more carbon

and may have greater levels of biodiversity than younger grasslands even if they are not truly “intact”. Clearly, the 25-year cutoff is imposed by data constraints rather than having an explicit ecological significance. I think it may be more effective to frame this comparison differently as “older” versus “younger” grassland as opposed to the idea of “potentially intact”, particularly since we do not really know the proportion of these older grasslands that have never been cultivated, and because the disturbance history of these grasslands likely varies among regions.

>>>Thanks for this feedback as well as the helpful specific suggestion to consider framing the comparison in terms of grassland age instead of potential intactness. We have heeded this recommendation and replaced the concept of “potentially intact” with “long-term” throughout the paper. This new term better matches how these grasslands are defined, is more candid about what the identified land represents, and also concedes that the land could have been cultivated and then converted to hay or pasture prior to the defined 25 years period.

Line 183: Following on the preceding comment, terms such as “likely intact” really don’t have a clear scientific meaning.

>>>Agreed. With the updated terminology, we have replaced associated phrases as well. Here, we replaced “were likely intact” with “met this criterion for longevity”.

Lines 207-238: I think I agree with most of the points made here in the discussion of yields, and I like the idea of exploring yield differentials at different scales (local and national). But I find myself questioning what the new insights are here. For me, it is already well understood that most of the prime agricultural land is already used for crops, and therefore agricultural expansion almost always has to occur in locations with lower yields. Perhaps including some more direct comparisons with results of previous agricultural land change studies would help to clarify how these results are extending our knowledge of agricultural land systems.

>>>We agree that the main findings seem as though they would already be well understood. However, in our review of the literature, we were not able to find many studies specifically observing these yield difference in the United States—and none that do so at field-level resolution for all locations across the conterminous US. A few global land use change modeling studies discuss and estimate yield differentials in relation to economic models, but the closest references to observed yield differentials that we could find came from Hendricks and Er (2018) and Lubowski et al (2006), where the authors looked at the subset of land moving into and out of the Conservation Reserve Program, and compare averages at the agricultural district and national levels. We have now added reference to these important previous studies to this part of the discussion: “[Our] field-level findings corroborate earlier estimates showing that the yields of land moving into and out of the CRP are lower than average for their agricultural district³⁷ and nationwide³⁸.”

Lines 314-316: Some additional information should be provided on the NLCD change estimates. I assumed that these estimates were derived from the newest (2016) version of the NLCD, which includes a 2008 as well as a 2016 epoch. But here the text implies that NLCD results are based on the 2001, 2006, and 2011 products. Please clarify to avoid confusion.

>>>Good catch. A citation indicating which NLCD products we used has now been added (in this first instance, it was the older generation of 2001, 2006, and 2011 products based on availability at the time of analysis). In addition, we have added the following sentence to the methods section *4.6 Accuracy estimation and comparison of results* to further clarify when we instead used the newer 2016 generation of NLCD products: “Note also that the version we compare to here (“NLCD 2016”)^{32,90} reflects a complete re-mapping of the NLCD series and is independent of the older generation of data incorporated into our trajectory-based land change analysis⁸³.”

Lines 418-422: I think that the estimation of conversion effects on milkweed stems was a weak link in the methodology. It is difficult to figure out how these estimates were generated without burrowing deeply into the supplemental materials. After reading the supplement, it is clear that these estimates are based on extrapolations from a relatively small dataset, with considerable (and unmeasurable) uncertainty as a result. I did think it was interesting to contrast the effects of grassland loss with those of GMOs and pesticides in croplands, and I don’t disagree with the idea that conversion is likely having a large impact on milkweeds and monarch habitat. But I believe that this results of this assessment need to be presented much more cautiously.

>>>Thanks for this helpful feedback and suggestion. We have now made several revisions to try to strengthen this component of the assessment: 1.) We added and now report an estimate of uncertainty (standard error) to the assessment of total milkweeds lost. 2.) We expanded the analysis to assess both the milkweeds lost on converted land and their relative losses compared to existing habitat, which better captures the overall impacts and provides additional insights that are more germane to our paper’s conclusions. 3.) We have added the following cautionary statement to the main text discussion and direct the readers to the supplementary materials which includes an expanded exploration of the uncertainty. “Our results embody significant uncertainty, however, and recent field surveys suggest milkweed concentrations of converted grasslands may be even greater than estimated here²⁷ (see **Supplementary note on milkweed conversion**).”

>>>The supplemental note is as follows:

“While our estimated milkweed loss numbers rely on the available data from the scientific literature, there remains substantial uncertainty in the magnitude of milkweed loss reported here. To help characterize this we calculated the standard errors for our estimates using reported error values for number of stems per acre from Pleasants (2016) and Thogmartin et al. (2017) for areas of wetlands, shrublands, CRP grasslands, and non-CRP grasslands based on the area of each from the estimates above^{2,5}. In addition to that associated with the assumed stem densities, there also remain other sources of uncertainty that are not captured as well as

variation in the representativeness of our data. For example, many of the milkweed stem density values were based on observations in Midwestern states located at the interior of the modeled region, and thus values for milkweed stems and losses may be more uncertain and variable around the periphery of the region⁵. In addition, our estimates account for only common milkweed (*Asclepias syriaca*), and thus total and relative loss of all milkweed species may vary. Although recent field surveys suggest common milkweeds outnumber the next most prevalent comparable variety by almost 10 to 1 in conservation grasslands in Minnesota, Wisconsin, and Iowa⁷, other areas like Kansas and Missouri have higher occurrences of less common species like *Asclepias viridis* and thus our estimates will be less representative of the changes occurring there⁸. Lastly, the estimates for milkweed stem densities are expected to vary widely from parcel to parcel and across landscape types, and thus the numbers presented here may represent overestimates in some areas and underestimates in others.”

Supplementary Figure 13: The performance of the yield models is impressive. I would be interested in seeing the relative importance of the predictor variables in the yield models for each crop. This information could be added in an extra figure or table in the supplementary material.

>>> Good idea. Variable importance plots for the yield models for each crop have now been added as Supplementary Figure 15.

Reviewer #3 (Remarks to the Author):

This manuscript tracked cropland expansion during 2008 to 2016 across CONUS using USDA NASS Cropland Data Layer. The authors then estimate the yield difference between expanded cropland and the benchmark yield, which would be either the national average or local average yield within a 10km window. The authors found that 69.5% of new cropland area produced yields lower than the national average, with a mean yield deficit of 6.5% (compared with national average). From those numbers, the authors argued that those new cropland produced “marginal” yields. On the other hand, the authors translated the cropland expansion area into loss of milkweed stems and nesting opportunities for waterfowl using existing methods. They found a loss of over 223 million milkweed stems in the Monarch butterfly's Midwest summer breeding range, and reduced waterfowl breeding habitat by over 138,000 nesting opportunities in the Prairie Pothole Region of the northern plains. They then argued that those are “high costs” to wildlife.

The manuscript is well-organized and well-written. However, I have the following concerns for the author to consider.

Firstly, I have a feeling that the definition of “marginal yields” and “high costs” are kind of arbitrary and subjective, especially when we are considering that the authors were comparing (or hinting a comparison between) some numbers of yield difference (which represent the crop productivity per unit area) with some other numbers of ecological cost (those are calculated for all the converted area). It's just like you are comparing yield with total production. I don't think any potential comparison like this is fair as they lack of similar benchmark. Also, we don't know the exact definition of “marginal yields” and “high costs”. I checked the numbers for yield difference, and found the numbers actually differed a lot between using national average yield and using local average yield as benchmark. The authors reported numbers in the abstract using national average yield as benchmark. However, we actually see much smaller number when using local average yield as benchmark. For this yield difference calculation, I think using local average yield is much more reasonable and would suggest the authors use the local average yield only. For the “high costs”, it seems there was not any benchmark. We actually don't know how high is really “high costs”.

>>>Thank you for this helpful feedback and identification of areas in need of revision and clarification. We agree that the use of “high costs” for the description of wildlife impacts was problematic and not appropriate terminology, as there was limited benchmark for comparison and also because we were mixing a relative metric (marginal yields) with an absolute metric (high costs). We have now updated the title to “disproportionate costs”, which better reflects the relative (rather than absolute) impact of cropland expansion on wildlife habitat identified in the study. We have also improved our analyses of the habitat impacts (detailed further below) to better support this conclusion that cropland expansion causes the loss of higher-than-average quality wildlife habitat.

>>>Regarding the reporting of yields, we have reviewed the abstract and believe that reporting the national (rather than local) yield differentials is still the most appropriate approach at this location, for two main reasons. First, for all other nationwide analyses (e.g. area of land conversion) we report the national-level results in the abstract. Reporting of the national-level yield differences therefore naturally follows and maintains proper consistency throughout. Second, interjection with the local yield differentials would require too much space to explain and defining the “local differential” concept falls beyond the scope of the abstract.

>>>With respect to the use of the term “marginal” in describing the ~6.5% yield deficit of new croplands, we believe this terminology is an ideal fit, but are open to alternative suggestions. The benefits of the term marginal are 1.) it is a relative term, and thus implies a comparison is being made (e.g. between new and existing croplands), and 2.) it provides indication of the degree of difference (in our case, a very small or narrow difference), which we believe is more appropriate than a general descriptor like “lower yields” which provides only an indication of direction but not of magnitude.

>>>While we do not directly define marginal yields, we introduce the term and discuss the concept throughout the discussion section with explicit reference to the lower yields of new croplands relative to existing cropland extent, and thereby feel that its meaning is clear and also justified to use in the title. Similarly, we use the term “disproportionate costs” in the discussion when summarizing the impacts to wildlife. Given the synoptic nature of these uses and terms, we think they are a good fit for the paper title.

Secondly, the authors used the method described in Pleasants (2016) to calculate the milkweeds lost, while Pleasants used the author’s earlier land cover change detection work (Lark et al., 2015). Compared to these two earlier works which used data from 2008 to 2012, this work used 4-year more data ranging from 2008 to 2016 with some technical revisions in the data processing workflow. This makes me feel that the innovation of this manuscript is reduced, though I admit that adding the waterfowl breeding habitat assessment component does increase the scope and novelty of this manuscript. Considering that CDL data is also available for 2017 and 2018, I am not sure why the authors excluded those two years.

>>>Thanks for the opportunity to clarify this. In addition to extending the duration of the Pleasants (2016) analysis, we improved upon its specificity to explicitly account for the conversion of conservation grasslands, which had a significant impact on the results—the *annual* loss of milkweed was 11-15x greater than that estimated by Pleasants (2016)—with broad implications for the significance of land use change relative to other factors affecting milkweed populations and monarch conservation.

>>>Regarding use of the CDL data, we do include the CDL data from 2017 (mentioned in what is now the first sentence of the 3rd paragraph of the methods section). However, data from the 2018 CDL was not released until February 2019 and was unavailable at the time we performed our analysis.

Thirdly, the ecological assessment lacks uncertainty quantification. I see there is some information uncertainty quantification in Pleasants’s work. Would it be possible that the authors conduct similar

work for uncertainty? This is important especially considering that those numbers are based on ecological survey and are prone to uncertainties. The authors should also add some discussions on the uncertainty in their estimations.

>>>Thanks for these great suggestions. We have now revised our ecological assessment to include an uncertainty estimate for the milkweed assessment based on the quantification in Pleasants's work. We have updated our results accordingly to report the standard error associated with our analysis, such that they now state "We estimate that approximately 220 (SE \pm 189) million common milkweed stems were lost on grasslands, wetlands, and shrublands converted to corn and soybean production across the Midwest during our study period".

>>>We have also added discussion of the uncertainty inherent to this analysis to the main text discussion section, with reference to the full paragraph about milkweed uncertainty in the supplemental. The newly added main text sentence is "Our results embody significant uncertainty, however, and recent field surveys suggest milkweed concentrations of converted grasslands may be even greater than estimated here²⁷ (see **Supplementary note on milkweed conversion**).". The full discussion of uncertainty in the supplemental note is copied in a response to a reviewer #2 comment above.

Other comments:

Yield modeling part: Could you please be more specific on the temporal and spatial resolutions of those predictors shown in supplementary Table 9? Did you use monthly value, yearly mean, or growing season mean values? And how did you aggregate those data into county level? All these details should be clarified.

>>>Thanks for this request. We have now greatly expanded the methods description of the yield model and now include a paragraph (copied below) describing the covariate layers, their spatial and temporal resolution, and how they were processed. Many of these details are also included in Supplementary Table 9 and its updated figure caption.

"Within each county we determined the biophysical characteristics associated with each of the yield observations by tabulating the mean values of gridded covariates (**Supplementary Table 9**) within the CDL-determined planted extent of each crop and year. Covariates included the mean, sum, minimum, and maximum values of multi-year (2008-2017) means of monthly gridded climate and water balance metrics from the TerraClimate database²⁵; the National Commodity Crop Productivity Index (NCCPI) for (i) corn/soybeans and (ii) small grains from the gSSURGO soils database⁸⁴; and elevation, slope and aspect grids derived from the National Elevation Database (NED)⁸⁵ in Google Earth Engine (GEE)⁸⁶. TerraClimate grids had a native spatial resolution of 2.5 arc minutes and were resampled to 30m prior to tabulation using the bilinear method in GEE to match the resolution of the CDL and the NCCPI grids. Grids derived from the NED had a 1/3 arc second resolution and were similarly aggregated in GEE to a 30m resolution. Tabulated covariate statistics were then joined to the corresponding yield data to complete the training set."

Reviewers' comments second round:

Reviewer #1 (Remarks to the Author):

The authors have done a great job responding to all three reviewers' comments and concerns in detail. The paper has substantially been revised and was greatly improved in my opinion. The results as well as the discussion and conclusions make a useful contribution to the literature.

Reviewer #2 (Remarks to the Author):

Overall, I think the revised manuscript is greatly improved compared to the original submission. The authors have done a very thorough job of responding to my comments and those from the other reviewers. I only have a few minor comments on the revision.

Line 44: Change ecosystems to ecosystems' or ecosystem.

Line 47: Not clear what "the region" is referring to here. Sounds like you are referring to the US, but the entire country is much larger than what we typically think of as a "region".

Line 48: Can you be more explicit about what you mean by "fine scale"? Do you mean at the level of individual crop fields?

Line 119: I would just say "forest" here, because you don't distinguish "timber land" as a separate class in your analysis.

Line 225: The wording here sounds a bit funny and conveys a lack of understanding of the underlying drivers of cropland expansion. Farmers don't expand cropland to achieve "aspirational production gains", they expand cropland to make money. I would suggest that most farmers know exactly what they are getting in to when they expand onto marginal lands, and do so only in situations where high crop prices make the lower yields profitable, or where crop insurance or other government subsidies reduce their risk.

Line 236: I would say we already know for certain that this is true. Farmers are already cropping the best croplands in just about every agricultural region. The only ways that they can increase production on the remaining lands would be through technological innovations or by planning new crops better suited to those lands.

Lines 256-258: This was an interesting finding, and I was surprised that there was little discussion. Was there regional variation, or was expansion onto hydric soils lower across the entire country? Does this finding reflect the effects of wetland protection policies? Or do the remaining areas with hydric soils tend to have very low potential for crop production?

Figure 2: This figure needs to be fixed up a bit. The state boundaries on the US map have a lot of gaps – I assume this is a problem with compression of the graphics file. It probably would help to make the boundaries a bit thicker. Also, the state names are very small and difficult to read, even though the geographic layout is helpful.

Reviewer #3 (Remarks to the Author):

I am glad that the authors addressed all my concerns and extensively revised their manuscript following comments from other two reviewers. I am now very happy to recommend this manuscript to be published in this journal. Great job!

Dear Reviewers,

Thank you very much for having taken the time to review our revised manuscript. Your original comments were extremely helpful in improving the work. We have made some additional minor revisions based on the suggestions of reviewer #2, as outlined below, and appreciate all of your kind words of support. We look forward to sharing our research with the scientific community.

REVIEWERS' COMMENTS:

Reviewer #1 (Remarks to the Author):

The authors have done a great job responding to all three reviewers' comments and concerns in detail. The paper has substantially been revised and was greatly improved in my opinion. The results as well as the discussion and conclusions make a useful contribution to the literature.

>>>Thank you for your support and for all of your previous helpful comments that improved the paper!

Reviewer #2 (Remarks to the Author):

Overall, I think the revised manuscript is greatly improved compared to the original submission. The authors have done a very thorough job of responding to my comments and those from the other reviewers. I only have a few minor comments on the revision.

>>>Thank you!

Line 44: Change ecosystems to ecosystems' or ecosystem.

>>>Good catch. We've updated this to "the integrity of ecosystems" for clarity.

Line 47: Not clear what "the region" is referring to here. Sounds like you are referring to the US, but the entire country is much larger than what we typically think of as a "region".

>>>Agreed. We've deleted "in the region" since it is unnecessary and introduces confusion.

Line 48: Can you be more explicit about what you mean by "fine scale"? Do you mean at the level of individual crop fields?

>>>Yes, and we have now updated this text to say "field scale."

Line 119: I would just say "forest" here, because you don't distinguish "timber land" as a separate class in your analysis.

>>>Updated accordingly.

Line 225: The wording here sounds a bit funny and conveys a lack of understanding of the underlying drivers of cropland expansion. Farmers don't expand cropland to achieve "aspirational production gains", they expand cropland to make money. I would suggest that most farmers know exactly what they are getting in to when they expand onto marginal lands, and do so only in situations where high crop prices make the lower yields profitable, or where crop insurance or other government subsidies reduce their risk.

>>> Good point. We've updated this sentence to remove the "aspirational" conjecture, such that it now states only the factual portion: "We found that croplands are moving onto lower-quality land in less-suitable regions—a dual setback to production gains from cropland expansion."

Line 236: I would say we already know for certain that this is true. Farmers are already cropping the best croplands in just about every agricultural region. The only ways that they can increase production on the remaining lands would be through technological innovations or by planning new crops better suited to those lands.

>>>Agreed. We've updated the language from "suggests" to "confirms" to reflect that this reinforces what we already know is true.

Lines 256-258: This was an interesting finding, and I was surprised that there was little discussion. Was there regional variation, or was expansion onto hydric soils lower across the entire country? Does this finding reflect the effects of wetland protection policies? Or do the remaining areas with hydric soils tend to have very low potential for crop production?

>>> We agree that this seems ripe for further exploration. We have added the following discussion about regional variation to Supplementary Figure 11: "Expansion frequently occurred on hydric soils throughout much of the Midwest region, especially in locations such as northern Minnesota and northeast Missouri, as well as in parts of the southeastern US and the Mississippi Alluvial Plain." We did not add discussion of the potential roles of wetland protection policy vs. the amount of remaining fertile hydric soils, as we would want to conduct a more in-depth assessment of these drivers before offering such an insight—though this seems to be an interesting question for future research.

Figure 2: This figure needs to be fixed up a bit. The state boundaries on the US map have a lot of gaps – I assume this is a problem with compression of the graphics file. It probably would help to make the boundaries a bit thicker. Also, the state names are very small and difficult to read, even though the geographic layout is helpful.

>>>Thanks for catching those issues. This indeed looks to have been a problem with the compression of the file in the word doc. In the high-resolution version, there are no state boundary gaps and the state names appear easier to read at full size. For supplementary Figure 7, which uses a similar figure layout

and format and will only be available as a doc/pdf, we have changed the orientation of the page to landscape, which allows for an enlarged image and easier reading.

Reviewer #3 (Remarks to the Author):

I am glad that the authors addressed all my concerns and extensively revised their manuscript following comments from other two reviewers. I am now very happy to recommend this manuscript to be published in this journal. Great job!

>>>Thank you very much! We appreciate the support and your previous suggestions that improved the manuscript.